# Better NTK Conditioning: A Free Lunch from (ReLU) Nonlinear Activation in Wide Neural Networks

**Chaoyue Liu**
Elmore Family School of Electrical and Computer Engineering
Purdue University
cyliu@purdue.edu

**Han Bi**
Department of Physics and Astronomy
Purdue University
bi53@purdue.edu

**Like Hui**
Halıcıoğlu Data Science Institute
University of California San Diego
lhui@ucsd.edu

**Xiao Liu**
Rosen Center For Advanced Computing
Purdue University
liu4201@purdue.edu

## Abstract

Nonlinear activation functions are widely recognized for enhancing the expressivity of neural networks, which is the primary reason for their widespread implementation. In this work, we focus on ReLU activation and reveal a novel and intriguing property of nonlinear activations. By comparing enabling and disabling the nonlinear activations in the neural network, we demonstrate their specific effects on wide neural networks: (a) *better feature separation*, i.e., a larger angle separation for similar data in the feature space of model gradient, and (b) *better NTK conditioning*, i.e., a smaller condition number of neural tangent kernel (NTK). Furthermore, we show that the network depth (i.e., with more nonlinear activation operations) further amplifies these effects; in addition, in the infinite-width-then-depth limit, all data are equally separated with a fixed angle in the model gradient feature space, regardless of how similar they are originally in the input space. Note that, without the nonlinear activation, i.e., in a linear neural network, the data separation remains the same as for the original inputs and NTK condition number is equivalent to the Gram matrix, regardless of the network depth. Due to the close connection between NTK condition number and convergence theories, our results imply that nonlinear activation helps to improve the worst-case convergence rates of gradient based methods.

## 1 Introduction

It is well known that nonlinear activation functions increase the expressivity of neural networks, which is the primary reason of their widespread implementation. A nonlinearly activated neural network can approximate any continuous function to arbitrary precision, as long as there are enough neurons in the hidden layers [13, 7, 12], while without it – as in a *linear neural network*, the network reduces to linear models of the input. In addition, deeper neural networks, which have more nonlinearly activated layers, have exponentially greater expressivity than shallower ones [32, 28, 29, 22, 34], indicating that the network depth promotes the power of nonlinear activation functions.

39th Conference on Neural Information Processing Systems (NeurIPS 2025).

In this paper, we reveal a novel and interesting effect of nonlinear activations that has not been previously noticed, despite their widespread application: the nonlinearity leads to larger data separation in the feature space of model gradient, and helps to decrease the condition number of neural tangent kernel (NTK). We also show that the depth of the network further amplifies these effects, namely, a deeper neural network has a better feature separation and a smaller NTK condition number, than a shallower one. While distinct and independent from the property of increasing expressivity, this property of nonlinear activations resembles the former in the manner that the effects vanish in the absence of nonlinear activations: removing the nonlinear activations in a neural network, the data separation and NTK condition number reduce to values observed in a linear model. Hence, the effect purely attributes to the presence of nonlinear activations.

Specifically, we first show the *better separation phenomenon*, i.e., improved separation for similar data in the model gradient feature space. We prove that, for a wide ReLU network $f$, any pair of data input vectors $\mathbf{x}$ and $\mathbf{z}$ that have similar directions (i.e., small but non-zero angle $\theta_{in}$ between $\mathbf{x}$ and $\mathbf{z}$) become more directionally separated in the model gradient space (i.e., model gradient angle $\phi$ between $\nabla f(\mathbf{x})$ and $\nabla f(\mathbf{z})$ is larger than $\theta_{in}$) with high probability of random initialization. We also find that deeper ReLU networks result in even better feature separation, i.e., larger $\phi$. Ultimately, in the infinite-width-then-depth limit, all data are equally separated with an angle $\sim 75.5°$ in the model gradient feature space, regardless of the input angle $\theta_{in}$, as long as $\theta_{in}$ is non-zero. Numerical simulation also show that the better separation phenomenon generalizes to other commonly used nonlinear activation functions, including GeLU, tanh, etc.

We further show the *better NTK conditioning* property of nonlinear activation, i.e., smaller NTK condition number. We prove that, as a consequence of the better feature separation, the NTK condition number of a wide ReLU network is strictly smaller than that without the nonlinearity, when the training dataset is not degenerate (i.e., no pair of training inputs are parallel). Moreover, with a larger depth, the NTK condition number becomes smaller. The intuition is that, if there exists a pair of similar inputs $\mathbf{x}$ and $\mathbf{z}$ in the training set (i.e., the angle between $\mathbf{x}$ and $\mathbf{z}$ is small), which is usually the case for large datasets, then NTK of linear neural networks must have close-to-zero smallest eigenvalues, resulting in extremely large NTK condition numbers. The activation makes these similar data more separated, hence it helps to increase the smallest eigenvalues of NTK, which in turn leads to a smaller NTK condition number. We further show that, in the infinite-width-then-depth limit, the NTK condition number of ReLU network converges to a fixed number $\frac{n+4}{3}$, which is independent of the data distribution and much smaller than typical NTK condition numbers.

*Connection with optimization theory.* While there could be multiple implications of the above property in various aspects, here we present its connection with existing optimization theories. Recent optimization theories showed that the NTK condition number $\kappa$, or the smallest eigenvalue of NTK, controls the theoretical convergence rate of gradient descent algorithms on wide neural networks [9, 8, 20]. Combined with these theories, our findings imply that: (a), the activation function has the effect of improving the worst-case convergence rate of gradient descent, and (b), deeper wide ReLU networks have faster convergence rate than shallower ones. Previous works often focus on accelerating convergence via a better function of $\kappa$ while assuming $\kappa$ is given and fixed. Our findings provide a different perspective of achieving acceleration by tuning $\kappa$ itself. Experimentally, we indeed find that deeper networks converge faster than shallower ones.

**Contributions.**   We summarize our contributions below. We find that:
- Nonlinear activation functions induce better separation between similar data in the feature space of model gradient. A larger network depth amplifies this better separation phenomenon.

- Nonlinear activations have the effect of decreasing the NTK condition number. A larger depth of the network further enhances this better NTK conditioning property.

- This better NTK conditioning property leads to faster convergence rate of gradient descent. We empirically verify this on various real world datasets.

The paper is organized as follow: Section 2 describes the setting and defines the key quantities and concepts, and analyzes linear neural networks as the baseline for comparison; Section 3 and 4 discuss our main results on the better separation and better conditioning of nonlinear activation, respectively; Section 5 discusses the implication on theoretical convergence rates; Section 6 concludes the paper. Proofs of theorems and main corollaries can be found in the appendix.

## 1.1 Related work

NTK and its spectrum have been extensively studied [19, 5, 21, 10, 11, 36, 25, 4, 6], since the discovery of constant NTK for infinitely wide neural networks [17]. [33] shows that the NTK spectrum of an infinitely wide ReLU network asymptotically exhibits a power law. Its distribution is further shown to be similar to that of Laplace kernel [11, 6], and can be computed [10]. Nguyen, Mondelli, and Montufar [25] analyzed the upper and lower bounds for the smallest NTK eigenvalue in $O()$ and $\Omega()$, respectively. With the assumption of spherically uniformly distributed data where the spectrum of (elementary-wise) power of the Gram matrix becomes simplified, [23], utilizing Hermite polynomials and power series expansion of NTK, provides the order of the smallest eigenvalue of the NTK of two-layer ReLU network in the infinite width limit. Under the same data setting, [3] computed the NTK eigenvalues for the two-layer ReLU network. Relying on the values of off-diagonal entries of the NTK matrix in the infinite *depth* limit, another work [36] analyzed the *asymptotic* dependence of the NTK condition number on the network depth $L$ for ReLU networks, which shows a decreasing trend as $L$ increases, consistent with our result.

In contrast to prior works, we are able to distill the effect of ReLU activation function via a sharp comparison between scenarios with and without ReLU, at any finite depth without data distribution assumption. Note that, without an assumption on data distribution, NTK spectral analysis becomes much harder and many data-distribution-dependent results may not hold any more. Moreover, at finite depth, off-diagonal entries of the NTK matrix has not converged and are typically quite different from its infinite depth limit, which makes analysis even harder.

We are aware of a prior work [2] which has results of similar flavor. It shows that the depth of a linear neural network may help to accelerate optimization via an implicit pre-conditioning of gradient descent. We note that this prior work is in an orthogonal direction, as its analysis is based on the linear neural network, which is activation-free, while our work focus on the better-conditioning effect of activation functions.

## 2 Setup and Preliminaries

**Notations for general purpose.** We denote the set $\{1, 2, \cdots, n\}$ by $[n]$. We use bold lowercase letters, e.g., $\mathbf{v}$, to denote vectors, and capital letters, e.g., $A$, to denote matrices. Given a vector, $\|\cdot\|$ denotes its Euclidean norm. Inner product between two vectors is denoted by $\langle \cdot, \cdot \rangle$. Given a matrix $A$, we denote its $i$-th row by $A_{i:}$, its $j$-th column by $A_{\cdot j}$, and its entry at $i$-th row and $j$-th column by $A_{ij}$. We also denote the expectation (over a distribution) of a variable by $\mathbb{E}[\cdot]$, and the probability of an event by $\mathbb{P}[\cdot]$. For a model $f(\mathbf{w}; \mathbf{x})$ which has parameters $\mathbf{w}$ and takes $\mathbf{x}$ as input, we use $\nabla f$ to denote its first derivative w.r.t. the parameters $\mathbf{w}$, i.e., $\nabla f := \partial f / \partial \mathbf{w}$.

**(Fully-connected) neural network.** Let $\mathbf{x} \in \mathbb{R}^d$ be the input, $m_l$ be the width (i.e., number of neurons) of the $l$-th layer, $W^{(l)} \in \mathbb{R}^{m_l \times m_{l-1}}$, $l \in [L+1]$, be the matrix of the parameters at layer $l$, and $\sigma(z)$ be the activation function, which is applied element-wise. A (fully-connected) neural network $f$, with $L$ hidden layers, is defined as:

$$
\begin{aligned}
\alpha^{(0)}(\mathbf{x}) &= \mathbf{x} \\
\alpha^{(l)}(\mathbf{x}) &= \frac{\sqrt{c_\sigma}}{\sqrt{m_l}} \sigma\left(W^{(l)} \alpha^{(l-1)}(\mathbf{x})\right), \quad \forall l \in \{1, 2, \cdots, L\}, \\
f(\mathbf{x}) &= W^{(L+1)} \alpha^{(L)}(\mathbf{x}),
\end{aligned}
\tag{1}
$$

where $c_\sigma = (\mathbb{E}_{z \sim \mathcal{N}(0,1)}[\sigma(z)^2])^{-1}$. For the special case of ReLU activation function: $\sigma(z) = \max\{0, z\}$, $c_\sigma = 2$. We also denote $\tilde{\alpha}^{(l)}(\mathbf{x}) \triangleq \frac{\sqrt{c_\sigma}}{\sqrt{m_l}} W^{(l)} \alpha^{(l-1)}(\mathbf{x})$. Following the NTK initialization scheme [17], these parameters are randomly initialized i.i.d. according to the normal distribution $\mathcal{N}(0, 1)$. The scaling factor $\sqrt{c_\sigma}/\sqrt{m_l}$ is introduced to normalize the hidden neurons [8]. We denote the collection of all parameters by $\mathbf{w}$.

**Remark 2.1.** *In this paper, we consider the bias-free setting where no bias term is included when computing the hidden neurons in Eq.(1). In fact, the bias term can potentially lead to different results, as have been noticed in [16].*

Without loss of generality, we set the layer widths as

$$m_0 = d, \ m_{L+1} = 1, \ and \ m_l = m, \ \forall \, l \in [L]. \tag{2}$$

and call $m$ as the network width. In the rest of the paper, we typically consider wide neural networks, i.e., networks with large widths $m$ and fixed depths $L$.

**Linear neural network.** For a comparison purpose, we also consider a linear neural network $\bar{f}$, which is the same as the neural network $f$ defined above, except that the activation function is the identity function $\sigma(z) = z$ and that the scaling factor of Eq.(1) is $1/\sqrt{m}$.

**Model gradient feature and neural tangent kernel (NTK).** Given a model $f$ (e.g., a neural network) with parameters $\mathbf{w}$, we call the the derivative of model $f$ with respect to all its parameters as the *model gradient feature* vector $\nabla f(\mathbf{w}; \mathbf{x})$ for the input $\mathbf{x}$. The NTK $\mathcal{K}$ is defined as

$$\mathcal{K}(\mathbf{w}; \mathbf{x}_1, \mathbf{x}_2) = \langle \nabla f(\mathbf{w}; \mathbf{x}_1), \nabla f(\mathbf{w}; \mathbf{x}_2) \rangle, \tag{3}$$

where $\mathbf{x}_1$ and $\mathbf{x}_2$ are two arbitrary network inputs. For a given dataset $\mathcal{D} = \{(\mathbf{x}_i, y_i)\}_{i=1}^n$, there is a gradient feature matrix $F$ such that each row $F_{i\cdot}(\mathbf{w}) = \nabla f(\mathbf{w}; \mathbf{x}_i)$ for all $i \in [n]$. The $n \times n$ NTK matrix $K(\mathbf{w})$ is defined such that its entry $K_{ij}(\mathbf{w})$, $i, j \in [n]$, is $\mathcal{K}(\mathbf{w}; \mathbf{x}_i, \mathbf{x}_j)$. It is easy to see that the NTK matrix

$$K(\mathbf{w}) = F(\mathbf{w})F(\mathbf{w})^T. \tag{4}$$

Note that the NTK for a linear model reduces to the Gram matrix $G \in \mathbb{R}^{d \times d}$, where each row of the matrix $X$ is an input feature $\mathbf{x}_i$, i.e., $X_{i\cdot} = \mathbf{x}_i^T$.

As pointed out by [21, 19, 17], a neural network with large width $m$ is approximately a linear model on the model gradient features $\nabla f(\mathbf{w}_0; \mathbf{x})$:

$$f(\mathbf{w}; \mathbf{x}) \approx f(\mathbf{w}_0; \mathbf{x}) + \nabla f(\mathbf{w}_0; \mathbf{x})^T (\mathbf{w} - \mathbf{w}_0) + O(1/\sqrt{m}). \tag{5}$$

Hence, the training dynamics of a wide neural network is largely controlled by the model gradient features of the training samples. We will see that the *model gradient angle*, i.e., the angle between the model gradient features of an arbitrary pair of inputs, is a key quantity that measures the mutual relations between training samples and is closely related to the NTK condition number and convergence rate.

**Definition 2.2** (Model gradient angle). *Given two arbitrary inputs $\mathbf{x}, \mathbf{z} \in \mathbb{R}^d$, define the model gradient angle as the angle between the model gradient vectors $\nabla f(\mathbf{x})$ and $\nabla f(\mathbf{z})$:*

$$\phi(\mathbf{x}, \mathbf{z}) \triangleq \arccos \left( \frac{\langle \nabla f(\mathbf{x}), \nabla f(\mathbf{z}) \rangle}{\|\nabla f(\mathbf{x})\| \|\nabla f(\mathbf{z})\|} \right).$$

**Condition number.** The *condition number* $\kappa$ of a positive definite matrix $A$ is defined as the ratio between its maximum eigenvalue and minimum eigenvalue:

$$\kappa = \lambda_{max}(A)/\lambda_{min}(A). \tag{6}$$

In the rest of the paper, we specifically denote the NTK matrix, NTK condition number and model gradient angle for the neural network as $K$, $\kappa$ and $\phi$, respectively, and denote their linear neural network counterparts as $\bar{K}$, $\bar{\kappa}$ and $\bar{\phi}$, respectively. We also denote the condition number of Gram matrix $G$ by $\kappa_0$.

## 2.1 Without nonlinear activation: the baseline for comparison

To distill the effect of the nonlinear activation function, we need a activation-free case as the baseline for comparison. This baseline is the linear neural network $\bar{f}$, with the same width and depth as $f$.

**Theorem 2.3.** *Consider a linear neural network $\bar{f}$. In the limit of infinite network width $m \to \infty$ and at network initialization $\mathbf{w}_0$, the following relations hold:*

- *for any input $\mathbf{x} \in \mathbb{R}^d$, $\|\nabla f(\mathbf{w}_0; \mathbf{x})\| = (L+1)\|\mathbf{x}\|$.*

- *for any inputs $\mathbf{x}, \mathbf{z} \in \mathbb{R}^d$, $\bar{\phi}(\mathbf{x}, \mathbf{z}) = \theta_{in}(\mathbf{x}, \mathbf{z})$.*

This theorem states that, without a nonlinear activation function, the model gradient map $\nabla f : \mathbf{x} \mapsto \nabla f(\mathbf{x})$ does not change the geometrical relationship between any data samples. For any input pairs, the model gradient angle $\bar{\phi}$ remains the same as the input angle $\theta_{in}$. Therefore, it is not surprising that the NTK of a linear network is the same as the Gram matrix (up to a constant factor), as formally stated in the following corollary (which can also be consistently obtained using Theorem 1 of [17]).

**Corollary 2.4** (NTK condition number without activation). *Consider a linear neural network $\bar{f}$. In the limit of infinite network width $m \to \infty$ and at network initialization, the NTK matrix $\bar{K} = (L+1)^2 G$. Moreover, $\bar{\kappa} = \kappa_0$.*

This corollary tells that, for a linear neural network, regardless of its depth $L$, the NTK condition number $\bar{\kappa}$ is always equal to the condition number $\kappa_0$ of the Gram matrix $G$. Therefore, any non-zero deviations, $\delta\phi \triangleq \phi - \theta_{in}$ from the input angle $\theta_{in}$, and $\delta\kappa \triangleq \kappa - \kappa_0$ from the Gram condition number $\kappa_0$, observed for a nonlinearly activated network $f$, should be attributed to the corresponding nonlinear activation.

# 3 Better separation in model gradient space

In this section, we show that the nonlinear activation function helps data separation in the model gradient space. Our theoretical analysis will focus on the special case of ReLU, and the results will be numerically verified on other nonlinear activations as well. Specifically, for two arbitrary inputs $\mathbf{x}$ and $\mathbf{z}$ with small $\theta_{in}(\mathbf{x}, \mathbf{z})$, we show that the model gradient angle $\phi(\mathbf{x}, \mathbf{z})$ is strictly larger than $\theta_{in}(\mathbf{x}, \mathbf{z})$, implying a better angle separation of the two data points in the model gradient space. Moreover, we show that the model gradient angle $\phi(\mathbf{x}, \mathbf{z})$ monotonically increases with the number of layers $L$, indicating that deeper network (more ReLU nonlinearity) has better angle separation.

First, we introduce an auxiliary quantity, *l-embedding angle* $\theta^{(l)}(\mathbf{x}, \mathbf{z})$, which measures the angle between two hidden vectors $\alpha^{(l)}(\mathbf{x})$ and $\alpha^{(l)}(\mathbf{z})$ at infinite width, and an auxiliary function $g : [0, \pi) \to [0, \pi)$ with $g(z) = \arccos\left(\frac{\pi-z}{\pi}\cos z + \frac{1}{\pi}\sin z\right)$. We also denote the $l$-fold composition of $g(\cdot)$ as $g^{\circ l}$. Please see Appendix A for the plot of the function and detailed discussion about its properties. As a highlight, $g$ has the following property: $g$ is approximately (but less than) the identity function $g(z) \approx z$ for small $z$, i.e., $z \ll 1$.

The following lemma gives the relation between the model gradient angle $\phi$ of any two inputs and their original input angle $\theta_{in}$, via the embedding angles $\theta^{(l)}$ and the function $g$.

**Lemma 3.1.** *Consider the ReLU network defined in Eq.(1) with L hidden layers and infinite network width. Given two arbitrary inputs $\mathbf{x}$ and $\mathbf{z}$, the angle $\phi(\mathbf{x}, \mathbf{z})$ between the model gradients $\nabla f(\mathbf{x})$ and $\nabla f(\mathbf{z})$ satisfies*

$$\cos\phi(\mathbf{x}, \mathbf{z}) = \frac{1}{L+1}\sum_{l=0}^{L}\left[\cos\theta^{(l)}(\mathbf{x}, \mathbf{z})\prod_{l'=l}^{L-1}(1 - \theta^{(l')}(\mathbf{x}, \mathbf{z})/\pi)\right] + O\left(\frac{1}{\sqrt{m}}\right), \quad (7)$$

*with $\theta^{(l)}(\mathbf{x}, \mathbf{z}) = g^{\circ l}(\theta_{in}(\mathbf{x}, \mathbf{z}))$. Moreover, $\|\nabla f(\mathbf{x})\| = \sqrt{L+1}\|\mathbf{x}\| + O\left(\frac{1}{\sqrt{m}}\right)$, for any $\mathbf{x}$.*

**Better feature separation.** Comparing with Theorem 2.3 for linear neural networks, we see that the nonlinear ReLU activation only affects the relative direction, but not the the magnitude, of the model gradient. Lemma 3.1 gives the relation between $\phi$ and the input angle $\theta_{in}$. Figure 1 plots $\phi$ as a function of $\theta_{in}$ for different network depth $L$.

The **key observation** is that: for relatively small input angles (say $\theta_{in} \le 30°$, which is actually not quite small), the model gradient angle $\phi$ is always greater than the input angle $\theta_{in}$. This suggests that, after the mapping $\nabla f : \mathbf{x} \mapsto \nabla f(\mathbf{x})$ from the input space to model gradient space, data inputs becomes more (directionally) separated, if they are similar in the input space (i.e., with small $\theta_{in}$). Comparing to the linear neural network case, where $\bar{\phi}(\mathbf{x}, \mathbf{z}) = \theta_{in}(\mathbf{x}, \mathbf{z})$ as in Theorem 2.3, we see that the ReLU nonlinearity results in a better angle separation $\phi(\mathbf{x}, \mathbf{z}) > \bar{\phi}(\mathbf{x}, \mathbf{z})$ for similar data.

Another observation is that: deeper ReLU networks lead to larger model gradient angles, when $\theta_{in} < 30°$. This indicates that deeper ReLU networks, which has more layers of ReLU nonlinear activation, makes the model gradient more separated between inputs. Note that, in the linear network case, the depth does not affect the gradient angle $\bar{\phi}$.

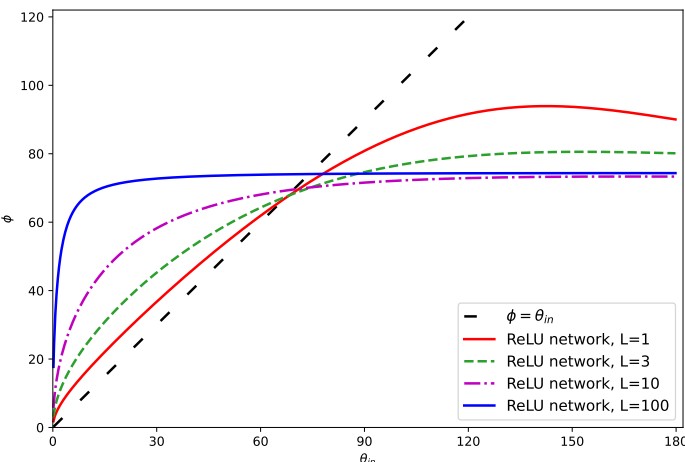

Figure 1: **Model gradient angles $\phi$ vs. input angle $\theta_{in}$ (according to Lemma 3.1).** Linear neural networks (black dash line), of any depth $L$, always have $\phi = \theta_{in}$. ReLU neural networks with various depths have better separation $\phi > \theta_{in}$ for similar data (i.e., small $\theta_{in}$). Deeper ReLU networks have better separation than shallow ones for similar data. All neural networks are infinitely wide.

Very similar inputs (i.e., when $\theta_{in}(\mathbf{x}, \mathbf{z}) \ll 1$), especially those with different labels, are often hard to distinguish and is one of the key factors that makes training difficult, because the decision boundary has to be fine-tuned to separate these very closely located inputs in order to make correct prediction. Hence, the regime of very small input angle ($\theta_{in} \ll 1$) is of particular interest for model training. The following theorem confirms the better separation in this regime.

**Theorem 3.2** (Better separation for similar data). *Consider two arbitrary inputs $\mathbf{x}, \mathbf{z} \in \mathbb{R}^d$, with small input angle $0 < \theta_{in}(\mathbf{x}, \mathbf{z}) \ll 1$, and the ReLU network defined in Eq.(1). The model angle $\phi(\mathbf{x}, \mathbf{z})$ is strictly greater than the input angle $\theta_{in}(\mathbf{x}, \mathbf{z})$:*

$$\phi(\mathbf{x}, \mathbf{z}) > \theta_{in}(\mathbf{x}, \mathbf{z}). \tag{8}$$

*with high probability of the random network initialization, if the network width $m = \Omega(1/\theta_{in}^2)$.*

The following corollary quantifies the better separation in this regime.

**Corollary 3.3.** *With the same setting as in Theorem 3.2 and with infinite width $m \to \infty$ but finite depth $L = \Omega(1/\theta_{in})$, $\cos\phi(\mathbf{x}, \mathbf{z}) = \left(1 - \frac{L}{2\pi}\theta_{in} + o(\theta_{in})\right)\cos\theta_{in}$.*

**Remark 3.4** (Separation in distance). *Indeed, the better angle separation discussed above implies a better separation in Euclidean distance as well. This can be easily seen by recalling from Lemma 3.1 that the model gradient mapping $\nabla f$ preserves the norm (up to a universal factor $L + 1$).*

We also point out that, Figure 1 indicates that for large input angles (say $\theta_{in} > 30°$) the model gradient angle $\phi$ is always large (greater than $30°$). Hence, non-similar data never become similar in the model gradient feature space.

**Better separation in infinite width and depth limit.** Now, we consider the infinite width and depth case. We took the infinite width limit a prior, this technically leads to the infinite-width-then-depth limit. The following theorem shows that, no matter how similar two inputs originally are, as long as they are not parallel, their model gradient features eventually get wide separated in large depth.

**Theorem 3.5.** *Consider the ReLU neural network defined in Eq.(1) and two non-parallel inputs $\mathbf{x}$ and $\mathbf{z}$, $\mathbf{x} \nparallel \mathbf{z}$. In the infinite-width-then-depth limit, the model gradient angle $\phi(\mathbf{x}, \mathbf{z})$ converges to a fix value $\arccos \frac{1}{4}$, regardless the input angle $\theta_{in}(\mathbf{x}, \mathbf{z})$.*

**Remark 3.6.** *The limit point value $\arccos \frac{1}{4}$ is about $75.5°$, which means the inputs are quite well-separated in the model gradient feature space, as network depth increase to infinity. Recall that, without the nonlinear activation, $\phi(\mathbf{x}, \mathbf{z}) = \theta_{in}$, which can be arbitrarily small.*

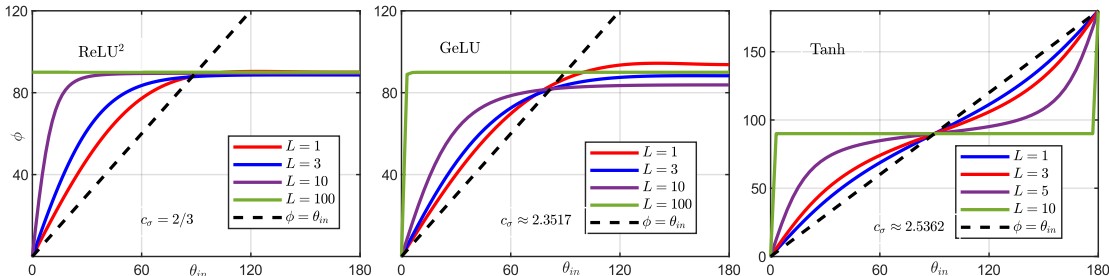

Figure 2: Better separation for non-ReLU activation functions. **Left**: ReLU$^2$, **Middle**: GeLU, **Right**: tanh. All plots are model gradient angle $\phi$ vs. input $\theta_{in}$.

It is interesting to observe that this limit point value is independent of the input angle, which means that the data points are mutually equally-separated in the limit. We will discuss its implications on NTK in the following sections.

**Beyond ReLU activation.**  Here, we numerically verify that the better separation phenomenon obtained for ReLU above also holds for other nonlinear activation functions. Figure 2 shows the relations between the model gradient angle $\phi$ and the input angle $\theta_{in}$ for the following nonlinear activations: ReLU$^2$ (i.e., $\sigma(z) = \max\{0, z^2\}$), GeLU and tanh. One can easily see that the better separation holds: for relatively small input angles $\theta_{in}$ (e.g., $\theta_{in} \leq 30°$), $\phi$ is always greater than $\theta_{in}$; and for deeper networks, $\phi$ is even greater. Interestingly, we observe that the gradient angle $\phi$ converges to $90°$ for these activation functions indicating gradient features become orthogonal in the limit of $L \to \infty$, different from the $75.5°$ that we obtain for ReLU networks.

We also show that the better separation generalizes beyond the NTK setting/regime. Please see Appendix B for more discussion.

## 4 Better NTK conditioning

In this section, we show both theoretically and experimentally that, the nonlinear activation induces a decrease in the NTK condition number $\kappa$. Moreover, a neural network with larger depth $L$, which means more nonlinear activations in operation, the NTK condition number $\kappa$ is generically smaller.

**Connection between condition number and model gradient angle.**  The smallest eigenvalue and condition number of NTK are closely related to the smallest model gradient angle $\min_{i,j \in [n]} \phi(\mathbf{x}_i, \mathbf{x}_j)$, through the gradient feature matrix $F$. Think about the case if $\phi(\mathbf{x}_i, \mathbf{x}_j) = 0$ (i.e., $\nabla f(\mathbf{x}_i)$ is parallel to $\nabla f(\mathbf{x}_j)$) for some $i, j \in [n]$, then $F$, hence NTK $K$, is not full rank and the smallest eigenvalue $\lambda_{min}(K)$ is zero, leading to an infinite condition number $\kappa$. Similarly, if $\min_{i,j \in [n]} \phi(\mathbf{x}_i, \mathbf{x}_j)$ is small, the smallest eigenvalue $\lambda_{min}(K)$ is also small, and condition number $\kappa$ is large, as stated in the following proposition (see proof in Appendix C).

**Proposition 4.1.** *Consider a $n \times n$ positive definite matrix $A = BB^T$, where matrix $B \in \mathbb{R}^{n \times d}$, with $d > n$, is of full row rank. Suppose that there exist $i, j \in [n]$ such that the angle $\phi$ between vectors $B_{i\cdot}$ and $B_{j\cdot}$ is small, i.e., $\phi \ll 1$, and that there exist constant $C > c > 0$ such that $c \leq \|B_{k\cdot}\| \leq C$ for all $k \in [n]$. Then, the smallest eigenvalue $\lambda_{min}(A) = O(\phi^2)$, and the condition number $\kappa = \Omega(1/\phi^2)$.*

Therefore, a good data angle separation in the model gradient features, i.e., $\min_{i,j \in [n]} \phi(\mathbf{x}_i, \mathbf{x}_j)$ not too small, is a necessary condition such that the condition number $\kappa$ is not too large. As is shown in the last section, the ReLU nonlinearity makes the samples more separated when mapped from the input data space to the model gradient feature space. Hence, it is expected that the NTK condition number will decrease in the presence of the ReLU nonlinearity.

**Smaller NTK condition number.**  Theoretically, we consider the infinite width limit. We require that the dataset is not degenerated, i.e., $\mathbf{x}_i \nparallel \mathbf{x}_j$ for all $i, j$. This is a mild and commonly used setting in the literature, see for example [9]. We require that the weights of the first layer $W^{(1)}$ be trainable

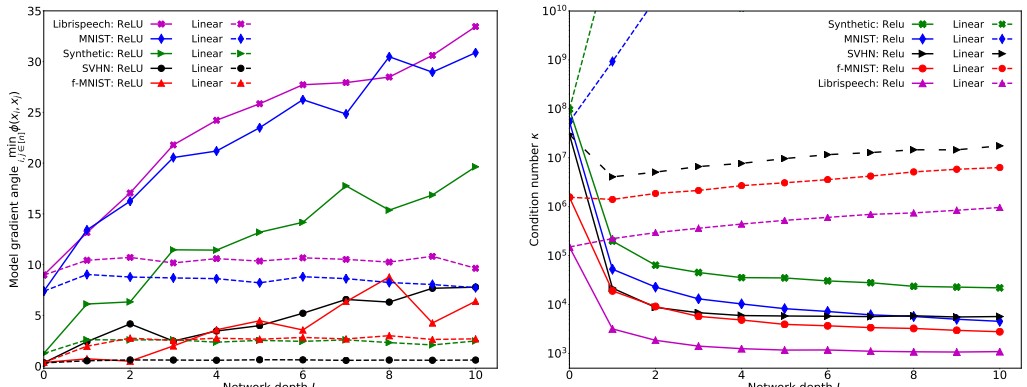

Figure 3: **Better separation (left) and Better NTK conditioning (right) of ReLU network** on various datasets. Solid lines are of ReLU networks, dashed lines are of linear neural networks for comparison. **Left:** Minimum $\phi$ (in degrees $^\circ$) vs. depth. ReLU network has better separation of model gradient feature as depth increases. **Right:** NTK condition number vs. depth. ReLU network has better conditioning of NTK as depth increases. Note that $L = 0$ corresponds to the case of a linear model and a linear neural network, and the NTK in this case is the Gram matrix.

and fix the other layers in the following theorem. This is also a common setting in literature to simplify the analysis [9].

**Theorem 4.2.** *Consider the ReLU network in Eq.(41) in the limit $m \to \infty$ and at initialization. Let the weights of the first layer $W^{(1)}$ be trainable and fix the other layers. We compare the two scenarios: (a) the network with ReLU activation and (b) the network with all the ReLU activation removed. The smallest eigenvalue $\lambda_{min}(K)$ of its NTK in scenario (a) is larger than that of scenario (b): $\lambda_{min}(K_a) > \lambda_{min}(K_b)$, and the NTK condition number $\kappa$ in scenario (a) is less than that in scenario (b): $\kappa_a < \kappa_b$. Moreover, for two ReLU neural networks $f_1$ of depth $L_1$ and $f_2$ of depth $L_2$ with $L_1 > L_2$, we have $\kappa_{f_1} < \kappa_{f_2}$.*

This theorem confirms the expectation that the NTK condition number $\kappa$ should be decreased, as a consequence of the existence of the ReLU nonlinearity. This theorem also shows that the depth of the ReLU network enhances this better NTK conditioning.

The high-level intuition behind the proof of this theorem is that: the derivative of the ReLU function, $\sigma'(z) = \mathbb{I}_{\{z \geq 0\}}$, resembles a binary gate which has *open* and *close* states. When ReLU are implemented, the model gradient map $\nabla f : \mathbf{x} \mapsto \nabla f(x)$ increases the directional diversity of the vectors $\nabla f(x)$, due to the high dimension of the model gradient space and the different activation patterns of the hidden layer for different samples $\mathbf{x}$. Hence, it is expected that the feature matrix $F$, as well as the NTK matrix $K$, is better conditioned.

In fact, fixing the weights of the top layer is not necessary and can be removed. We relax this requirement in Appendix F. In our experiments in Section 4.1 where all layers are trainable, we observe the phenomena of *better separation* and *better NTK conditioning*.

**NTK condition number in infinite depth.** As a consequence of the pairwise equal-separation result (Theorem 3.5), the NTK matrix got simplified in the infinite depth limit. The following theorem shows that the NTK condition number converges to a fixed value $\frac{n+4}{3}$, which is independent of the data distribution.

**Theorem 4.3.** *Consider the ReLU neural network defined in Eq.(1) and a dataset $\mathcal{D} = \{(\mathbf{x}_i, y_i)\}_{i=1}^n$. Suppose that all data inputs are normalized $\|\mathbf{x}_i\| = 1$ for all $i$, and $\mathbf{x}_i \nparallel \mathbf{x}_j$ for all $i \neq j$. In the infinite-width-then-depth limit, the NTK condition number $\kappa$ converges to $\frac{n+4}{3}$.*

### 4.1 Experimental evidence

Here, we experimentally show that better separation and better conditioning happen in practice.

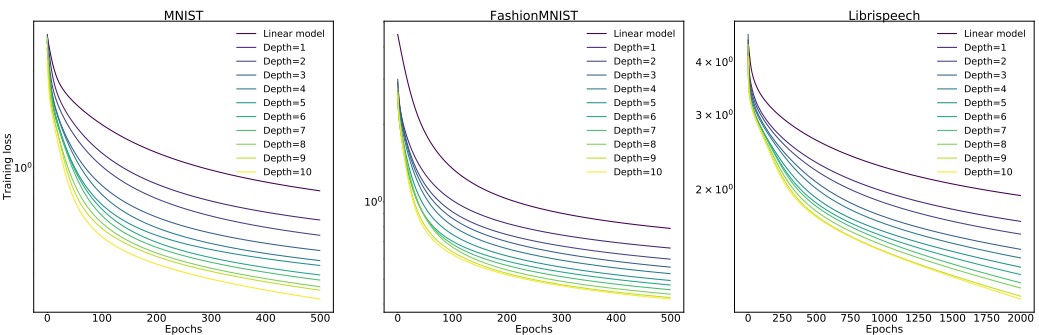

Figure 4: **Training curve of ReLU networks with different depths.** On each of these datasets, we see that deeper ReLU network always converges faster than shallower ones.

**Dataset.** We use the following datasets: synthetic dataset, MNIST [18], FashionMNIST (f-MNIST) [35], SVHN [24] and Librispeech [27]. The synthetic data consists of 2000 samples which are randomly drawn from a 5-dimensional Gaussian distribution with zero-mean and unit variance. The MNIST, f-MNIST and SVHN datasets are image datasets where each input is an image. The Librispeech is a speech dataset including 100 hours of clean speeches. In the experiments, we use a subset of Librispeech with $50,000$ samples, and each input is a 768-dimensional vector representing a frame of speech audio and we follow [15] for the feature extraction.

**Models.** For each of the datasets, we use a ReLU activated fully-connected neural network architecture to process. The ReLU network has $L$ hidden layers, and has $512$ neurons in each of its hidden layers. The ReLU network uses the NTK parameterization and initialization strategy (see [17]). For each dataset, we vary the network depth $L$ from 0 to 10. Note that $L = 0$ corresponding to the linear model case. In addition, for comparison, we use a linear neural network, which has the same architecture with the ReLU network except the absence of activation function.

**Results.** For each dataset and given network depth $L$, we evaluate both the smallest pairwise model gradient angle $\min_{i,j \in [n]} \phi(\mathbf{x}_i, \mathbf{x}_j)$ and the NTK condition number $\kappa$, at the network initialization. We take 5 independent runs over 5 random initialization seeds, and report the average. In each run, we used a A-100 GPU to compute the NTK, which took $4 \sim 10$ hours. The results are shown in Figure 3. We compare the two scenarios of *with* and *without* the ReLU activation function. As one can easily see from the plots, a ReLU network (depth $L = 1, 2, \cdots, 10$) always have a better separation of data features (i.e., larger smallest pairwise model gradient angle), and a better NTK conditioning (i.e., smaller NTK condition number), than its corresponding linear network (compare the solid line and dash line of the same color). Furthermore, the monotonically decreasing NTK condition number shows that a deeper ReLU network have a better conditioning of NTK.

## 5 Optimization acceleration

Recently studies have shown strong connections between the NTK condition number and the theoretical convergence rate of gradient descent algorithms on wide neural networks [9, 8, 31, 1, 37, 26, 20]. In [9, 8, 20], the worst-case convergence rate has been shown to be

$$L(\mathbf{w}_t) \leq (1 - \kappa^{-1})^t L(\mathbf{w}_0). \tag{9}$$

Although $\kappa$ is evaluated on the entire optimization path, all these theories used the fact that NTK is almost constant for wide neural networks and an evaluation at initialization $\mathbf{w}_0$ is enough.

As a smaller NTK condition number (or larger smallest eigenvalue of NTK) implies a faster worst-case convergence rate, our findings suggest that: (a), the ReLU activation function helps improve the worst-case convergence rate of gradient descent, and (b), deeper wide ReLU networks have faster convergence rate than shallower ones.

We experimentally verify this implication. Specifically, we train the ReLU networks, with depth $L$ ranging from 1 to 10, for the datasets MNIST, f-MNIST, and Librispeech. For all training tasks, we use cross-entropy loss as the objective function and use mini-batch stochastic gradient descent (SGD)

of batch size 500 to optimize. For each task, we find its optimal learning rate by grid search. On MNIST and f-MNIST, we train 500 epochs, and on Librispeech, we training 2000 epochs.

The curves of training loss against epochs are shown in Figure 4. We observe that, for all these datasets, a deeper ReLU network always converges faster than a shallower one. This is consistent with the theoretical prediction that the deeper ReLU network, which has a smaller NTK condition number, has a faster theoretical convergence rate.

**Trade-off between optimization and generalization.** Although a faster convergence in terms of number of iterations for deep networks, as Theorem 3.5 suggests, in the extreme case of infinite depth $L \to \infty$, any non-parallel input pairs become equally separated in gradient features regardless of their original similarity. Even though not mutually orthogonal, this could also result in a trivial generalization: close to random guess for unseen data. The same consequence can also be obtained from [14], where they dropped the initial random guess value and obtained a zero prediction for unseen data.

As for finite depth, it is theoretically hard to predict at what depth this trade-off starts to happen. Under the same experimental setting as in Figure 4, Table 1 shows that the generalization performance starts to decrease at depth $L = 8$, suggesting a optimization-generalization trade-off for large depth.

Table 1: **Generalization dependence on ReLU network depth** $L$**.** Test accuracies are reported after training convergence on MNIST.

| Depth $L$ | 1 | 3 | 6 | 8 | 10 | 12 |
|---|---|---|---|---|---|---|
| test accuracy $(\%)$ | 95.98 | 97.43 | 97.57 | 97.52 | 97.39 | 97.19 |

## 6    Conclusion and discussions

In this work, we showed the effects of nonlinear activation on better separation of similar data in feature space and on the NTK conditioning. We also showed that more sequential activation operations, i.e., larger network depth, amplifies these effects. As the NTK conditioning is closely related to theoretical convergence rate of gradient descent, our findings also suggest a positive role of activation functions in optimization theories. A limitation of the paper is that the theoretical analysis is only conducted on ReLU activation, although results have been empirically verified for other nonlinear activations. For other activations, the analysis requires analytical expressions for integrations involved, which requires a distinct type of analysis and we consider it as a future work.

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

# A    Properties of function $g$

Recall that the function $g : [0, \pi) \to [0, \pi)$ is defined as (see Lemma E.5)

$$g(z) = \arccos\left(\frac{\pi - z}{\pi}\cos z + \frac{1}{\pi}\sin z\right), \tag{10}$$

Figure 5 shows the plot of this function. From the plot, we can easily find the following properties.

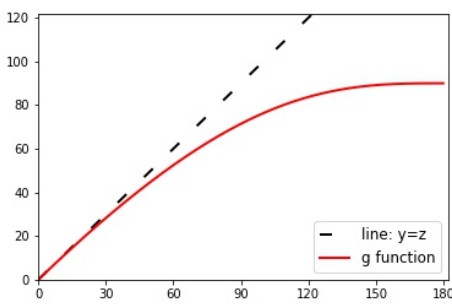

Figure 5: Curve of the function $g(\theta)$. As can be seen, $g(\theta)$ is monotonic, and is approximately the identity function $y = \theta$ in the small angle region ($\theta \ll 90°$).

**Proposition A.1** (Properties of $g$). *The function $g$ defined in Eq.(10) has the following properties:*

1. *$g$ is a monotonically increasing function;*

2. *$g(z) \leq z$, for all $z \in [0, \pi)$; and $g(z) = z$ if and only if $z = 0$;*

3. *for any $z \in [0, \pi)$, the sequence $\{g^l(z)\}_{l=1}^{\infty}$ is monotonically decreasing, and has the limit $\lim_{l \to \infty} g^l(z) = 0$.*

*Proof.*  **Part 1.** First, we consider the auxiliary function $\tilde{g}(z) = \frac{\pi - z}{\pi}\cos z + \frac{1}{\pi}\sin z$. We see that

$$\frac{d\tilde{g}(z)}{dz} = -\left(1 - \frac{z}{\pi}\right)\sin z \leq 0, \ \ \forall z \in [0, \pi).$$

Hence, $\tilde{g}(z)$ is monotonically decreasing on $[0, \pi)$. Combining with the monotonically decreasing nature of the $\arccos$ function, we get that $g$ is monotonically increasing.

**Part 2.** It suffices to prove that $\cos z \leq \tilde{g}(z)$ and that the equality holds only at $z = 0$. For $z = 0$, it is easy to check that $\cos z = \tilde{g}(z)$, as both $z$ and $\sin z$ are zero. For $z \in (0, \pi/2)$, noting that $\tan z - z > 0$, we have

$$\tilde{g}(z) = \frac{\pi - z}{\pi}\cos z + \frac{1}{\pi}\sin z = \cos z + \frac{1}{\pi}\left(-z + \tan z\right)\cos z > \cos z. \tag{11}$$

For $z = \pi/2$, we have $\cos\pi/2 = 0 < 1/\pi = \tilde{g}(\pi/2)$. For $z \in (\pi/2, \pi)$, we have the same relation as in Eq.(11). The only differences are that, in this case, $\cos z < 0$ and $\tan z - z < 0$. Therefore, we still get $\tilde{g}(z) > \cos z$ for $z \in (\pi/2, \pi)$.

**Part 3.** From part 2, we see that $g(z) < z$ for all $z \in (0, \pi)$. Hence, for any $l$, $g^{l+1}(z) < g^l(z)$. Moreover, since $z = 0$ is the only fixed point such that $g(z) = z$, in the limit $l \to \infty$, $g^l(z) \to 0$.  $\square$

It is worth to note that the last property of $g$ function immediately implies the collapse of embedding vectors from different inputs in the infinite depth limit $L \to \infty$. This embedding collapse has been observed in prior works [28, 30] (although by different type of analysis) and has been widely discussed in the literature of Edge of Chaos.

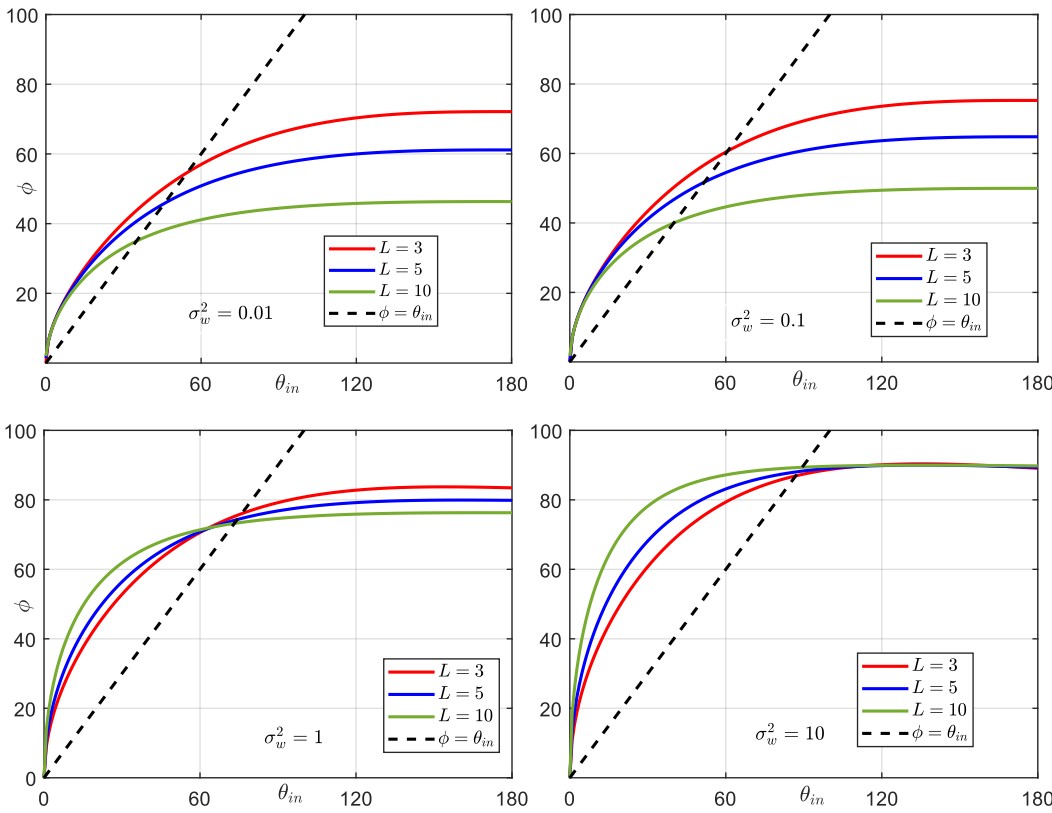

Figure 6: Model gradient angle $\phi$ vs. input $\theta_{in}$ in different scaling regimes $\sigma_{\mathbf{w}}^2 = 0.01, 0.1, 1$ and 10. The better feature separation always holds for similar data (when $\theta_{in}$ is small, left end of each plot).

**Theorem A.2.** *Consider a ReLU neural network. Given any two inputs $\mathbf{x}, \mathbf{z} \in \mathbb{R}^d$, the sequence of angles $\{\theta^{(l)}(\mathbf{x}, \mathbf{z})\}_{l=1}^L$ between their $l$-embedding vectors $\alpha^{(l)}(\mathbf{x})$ and $\alpha^{(l)}(\mathbf{z})$, is monotonically decreasing. Moreover, in the limit of infinite depth,*

$$\lim_{L \to \infty} \theta^{(L)}(\mathbf{x}, \mathbf{z}) = 0, \tag{12}$$

*and there exists a vector $\alpha$ such that, for any input $\mathbf{x}$, the last layer $L$-embedding*

$$\alpha^{(L)}(\mathbf{x}) = \|\mathbf{x}\|\alpha. \tag{13}$$

## B  Beyond the NTK regime

Here, we show that the better separation phenomenon still holds outside of the NTK regime. We consider different initialization scales $\mathbf{w} \sim \mathcal{N}(0, \sigma_{\mathbf{w}}^2)$. Note that $\sigma_{\mathbf{w}}^2 < 1$ corresponds to small initialization. Figure 6 plots the model gradient angle $\phi$ as a function of the input $\theta_{in}$, for different scaling regimes: $\sigma_{\mathbf{w}}^2 = 0.01, 0.1, 1$ and 10. It shows that the better separation phenomenon still holds for similar inputs at various network depths.

## C  Proof of Proposition 4.1

*Proof.* Consider the matrix $B$ and the $n$ vectors $\mathbf{b}_k \triangleq B_{k\cdot}$, $k \in [n]$. The smallest singular value square of matrix $B$ is defined as

$$\sigma_{min}^2(B) = \min_{\mathbf{v} \neq 0} \frac{\mathbf{v}^T B B^T \mathbf{v}}{\mathbf{v}^T \mathbf{v}} = \min_{\mathbf{v} \neq 0} \frac{\|\sum_k v_k \mathbf{b}_k\|^2}{\|\mathbf{v}\|^2}.$$

Since the angle $\phi$ between $\mathbf{b}_i = B_i$. and $\mathbf{b}_j = B_j$. is small, let $\mathbf{v}'$ be the vector such that $v'_i = \|\mathbf{b}_j\|$, $v'_j = -\|\mathbf{b}_i\|$ and $v'_k = 0$ for all $k \neq i, j$. Then

$$
\sigma^2_{min}(B) \leq \frac{\|\sum_k v'_k \mathbf{b}_k\|^2}{\|\mathbf{v}'\|^2} = \left\| \frac{\|\mathbf{b}_j\|}{\sqrt{\|\mathbf{b}_i\|^2 + \|\mathbf{b}_j\|^2}} \mathbf{b}_i - \frac{\|\mathbf{b}_i\|}{\sqrt{\|\mathbf{b}_i\|^2 + \|\mathbf{b}_j\|^2}} \mathbf{b}_j \right\|^2
$$

$$
= \frac{2\|\mathbf{b}_i\|^2\|\mathbf{b}_j\|^2}{\|\mathbf{b}_i\|^2 + \|\mathbf{b}_j\|^2}(1 - \cos\phi)
$$

$$
= \frac{\|\mathbf{b}_i\|^2\|\mathbf{b}_j\|^2}{\|\mathbf{b}_i\|^2 + \|\mathbf{b}_j\|^2}\phi^2 + O(\phi^4).
$$

Since $A = BB^T$, the smallest eigenvalue $\lambda_{min}(A)$ of $A$ is the same as $\sigma^2_{min}(B)$.

On the other hand, the largest eigenvalue $\lambda_{max}(A)$ of matrix $A$ is lower bounded by $\mathrm{tr}(A)/n$. Note that the diagonal entries $A_{kk} = \|\mathbf{b}_k\|$. Hence, $c \leq \lambda_{max}(A) \leq C$. Therefore, the condition number $\kappa = \lambda_{max}(A)/\lambda_{min}(A) = \Omega(1/\phi^2)$. $\qquad\square$

# D Proofs of Theorems without (ReLU) activation

## D.1 Proof of Theorem 2.3

*Proof.* First of all, we provide a useful lemma.

**Lemma D.1.** *Consider a matrix $A \in \mathbb{R}^{m \times d}$, with each entry of $A$ is i.i.d. drawn from $\mathcal{N}(0, 1)$. In the limit of $m \to \infty$,*

$$
\frac{1}{m}A^T A \to I_{d \times d}, \quad \textit{in probability.} \tag{14}
$$

We first consider the embedding vectors $\bar{\alpha}^{(l)}$ and the embedding angles $\bar{\theta}^{(l)}$. By definition of linear neural network, we have, for all $l \in [L]$ and input $\mathbf{x} \in \mathbb{R}^d$,

$$
\bar{\alpha}^{(l)}(\mathbf{x}) = \frac{1}{m^{l/2}}W^{(l)}W^{(l-1)}\cdots W^{(1)}\mathbf{x}. \tag{15}
$$

Note that at the network initialization entries of $W^{(l)}$ are i.i.d. and follows $\mathcal{N}(0, 1)$. Hence, the inner product

$$
\langle \bar{\alpha}^{(l)}(\mathbf{x}), \bar{\alpha}^{(l)}(\mathbf{z}) \rangle = \frac{1}{m^l}\mathbf{x}^T W^{(1)T}\cdots W^{(l-1)T}W^{(l)T}W^{(l)}W^{(l-1)}\cdots W^{(1)}\mathbf{z} \overset{(a)}{=} \mathbf{x}^T\mathbf{z},
$$

where in step (a) we recursively applied Lemma D.1 $l$ times. Putting $\mathbf{z} = \mathbf{x}$, we get $\|\bar{\alpha}^{(l)}(\mathbf{x})\| = \|\mathbf{x}\|$, for all $l \in [L]$. By the definition of embedding angles, it is easy to check that $\bar{\theta}^{(l)}(\mathbf{x}, \mathbf{z}) = \theta_{in}(\mathbf{x}, \mathbf{z})$, for all $l \in [L]$.

Now, we consider the model gradient $\nabla\bar{f}$ and the model gradient angle $\bar{\phi}$. As we consider the model gradient only at network initialization, we don't explicitly write out the dependence on $\mathbf{w}_0$, and we write $\nabla\bar{f}(\mathbf{w}_0, \mathbf{x})$ simply as $\nabla\bar{f}(\mathbf{x})$. The model gradient $\nabla\bar{f}$ can be decomposed as

$$
\nabla\bar{f}(\mathbf{x}) = (\nabla_1\bar{f}(\mathbf{x}), \nabla_2\bar{f}(\mathbf{x}), \cdots, \nabla_{L+1}\bar{f}(\mathbf{x})), \quad \textit{with } \nabla_l\bar{f}(\mathbf{x}) = \frac{\partial\bar{f}(\mathbf{x})}{\partial W^{(l)}}, \forall l \in [L+1]. \tag{16}
$$

Hence, the inner product

$$
\langle \nabla\bar{f}(\mathbf{x}), \nabla\bar{f}(\mathbf{z}) \rangle = \sum_{l=1}^{L+1}\langle \nabla_l\bar{f}(\mathbf{x}), \nabla_l\bar{f}(\mathbf{z}) \rangle,
$$

and for all $l \in [l+1]$,

$$
\langle \nabla_l\bar{f}(\mathbf{x}), \nabla_l\bar{f}(\mathbf{z}) \rangle = \langle \bar{\alpha}^{(l-1)}(\mathbf{x}), \bar{\alpha}^{(l-1)}(\mathbf{z}) \rangle \cdot \langle \prod_{l'=l+1}^{L+1}\frac{1}{\sqrt{m}}W^{(l')T}, \prod_{l'=l+1}^{L+1}\frac{1}{\sqrt{m}}W^{(l')T} \rangle \overset{(b)}{=} \mathbf{x}^T\mathbf{z}.
$$

Here in step (b), we again applied Lemma D.1. Therefore,

$$
\langle \nabla\bar{f}(\mathbf{x}), \nabla\bar{f}(\mathbf{z}) \rangle = (L+1)\mathbf{x}^T\mathbf{z}. \tag{17}
$$

Putting $\mathbf{z} = \mathbf{x}$, we get $\|\nabla f(\mathbf{x})\| = (L+1)\|\mathbf{x}\|$. By the definition of model gradient angle, it is easy to check that $\bar{\phi}(\mathbf{x}, \mathbf{z}) = \theta_{in}(\mathbf{x}, \mathbf{z})$. $\qquad\square$

# E   Proofs of Theorems for ReLU network

## E.1   Preliminary results

Before the proofs, we introduce some useful notations and lemmas. The proofs of these lemmas are deferred to Appendix G.

Given a vector $\mathbf{v} \in \mathbb{R}^p$, we define the following diagonal indicator matrix:

$$\mathbb{I}_{\{\mathbf{v} \geq 0\}} = \mathsf{diag}\left(\mathbb{I}_{\{v_1 \geq 0\}}, \mathbb{I}_{\{v_2 \geq 0\}}, \cdots, \mathbb{I}_{\{v_p \geq 0\}}\right), \tag{18}$$

with

$$\mathbb{I}_{\{v_i \geq 0\}} = \left\{ \begin{array}{ll} 1 & v_i \geq 0, \\ 0 & v_i < 0. \end{array} \right.$$

**Lemma E.1.** *Consider two vectors* $\mathbf{v}_1, \mathbf{v}_2 \in \mathbb{R}^p$ *and a p-dimensional random vector* $\mathbf{w} \sim \mathcal{N}(0, I_{p \times p})$. *Denote* $\theta$ *as the angle between* $\mathbf{v}_1$ *and* $\mathbf{v}_2$, *i.e.,* $\cos \theta = \frac{\langle \mathbf{v}_1, \mathbf{v}_2 \rangle}{\|\mathbf{v}_1\|\|\mathbf{v}_2\|}$. *Then, the probability*

$$\mathbb{P}[(\mathbf{w}^T \mathbf{v}_1 \geq 0) \wedge (\mathbf{w}^T \mathbf{v}_2 \geq 0)] = \frac{1}{2} - \frac{\theta}{2\pi}. \tag{19}$$

**Lemma E.2.** *Consider two arbitrary vectors* $\mathbf{v}_1, \mathbf{v}_2 \in \mathbb{R}^p$ *and a random matrix* $W \in \mathbb{R}^{q \times p}$ *with entries* $W_{ij}$ *i.i.d. drawn from* $\mathcal{N}(0, 1)$. *Denote* $\theta$ *as the angle between* $\mathbf{v}_1$ *and* $\mathbf{v}_2$, *and define* $\mathbf{u}_1 = \frac{\sqrt{2}}{\sqrt{q}} \sigma(W \mathbf{v}_1)$ *and* $\mathbf{u}_2 = \frac{\sqrt{2}}{\sqrt{q}} \sigma(W \mathbf{v}_2)$. *Then, in the limit of* $q \to \infty$,

$$\langle \mathbf{u}_1, \mathbf{u}_2 \rangle = \frac{1}{\pi} \left( (\pi - \theta) \cos \theta + \sin \theta \right) \|\mathbf{v}_1\|\|\mathbf{v}_2\|. \tag{20}$$

**Lemma E.3.** *Consider two arbitrary vectors* $\mathbf{v}_1, \mathbf{v}_2 \in \mathbb{R}^p$ *and two random matrices* $U \in \mathbb{R}^{s \times q}$ *and* $W \in \mathbb{R}^{q \times p}$, *where all entries* $U_{ij}$, $i \in [s]$ *and* $j \in [q]$, *and* $W_{kl}$, $k \in [q]$ *and* $l \in [p]$, *are i.i.d. drawn from* $\mathcal{N}(0, 1)$. *Denote* $\theta$ *as the angle between* $\mathbf{v}_1$ *and* $\mathbf{v}_2$, *and define matrices* $A_1 = \frac{\sqrt{2}}{\sqrt{q}} U \mathbb{I}_{\{W \mathbf{v}_1 \geq 0\}}$ *and* $A_2 = \frac{\sqrt{2}}{\sqrt{q}} U \mathbb{I}_{\{W \mathbf{v}_2 \geq 0\}}$. *Then, in the limit of* $q \to \infty$, *the matrix*

$$A_1 A_2^T = \frac{\pi - \theta}{\pi} I_{s \times s}. \tag{21}$$

**Lemma E.4.** *Consider matrix* $B = A A^T$ *with* $A \in \mathbb{R}^{n \times p}$ *and a random matrix* $W \in \mathbb{R}^{q \times p}$ *where all entries of* $W$ *are i.i.d. drawn from* $\mathcal{N}(0, 1)$. *Define the tensor* $\mathbf{A}' \in \mathbb{R}^{n \times p \times q}$, *such that* $\mathbf{A}'_{ikl} := \sqrt{2} A_{ik} \mathbb{I}_{\{W_{l:} A_{i:} \geq 0\}}$. *Let* $B' \in \mathbb{R}^{n \times n}$ *be the matrix such that each entry* $B'_{ij} = \sum_{k,l} \mathbf{A}'_{ikl} \mathbf{A}'_{jkl}$. *Then, in the limit of* $q \to \infty$, *the smallest and largest eigenvalues satisfy:* $\lambda_{min}(B') > \lambda_{min}(B)$, *and* $\lambda_{max}(B') < \lambda_{max}(B)$.

## E.2   Proof of Lemma 3.1

*Proof.* The model gradient $\nabla f(\mathbf{x})$ is composed of the components $\nabla_l f(\mathbf{x}) \triangleq \frac{\partial f}{\partial W^l}$, for $l \in [L + 1]$. Each such component has the following expression: for $l \in [L + 1]$

$$\nabla_l f(\mathbf{x}) = \alpha^{(l-1)}(\mathbf{x}) \delta^{(l)}(\mathbf{x}), \tag{22}$$

where

$$\delta^{(l)}(\mathbf{x}) = \left(\frac{2}{m}\right)^{\frac{L-l+1}{2}} W^{(L+1)} \mathbb{I}_{\{\tilde{\alpha}^{(L)}(\mathbf{x}) \geq 0\}} W^{(L)} \mathbb{I}_{\{\tilde{\alpha}^{(L-1)}(\mathbf{x}) \geq 0\}} \cdots W^{(l+1)} \mathbb{I}_{\{\tilde{\alpha}^{(l)}(\mathbf{x}) \geq 0\}}. \tag{23}$$

Note that in Eq.(22), $\nabla_l f(\mathbf{x})$ is an outer product of a column vector $\alpha^{(l-1)}(\mathbf{x}) \in \mathbb{R}^{m_{l-1} \times 1}$ ($m_{l-1} = d$ if $l = 1$, and $m_{l-1} = m$ otherwise) and a row vector $\delta^{(l)}(\mathbf{x}) \in \mathbb{R}^{1 \times m_l}$ ($m_l = 1$ if $l = L + 1$, and $m_l = m$ otherwise).

First, we consider an infinitely wide neural network $f^\infty$ of depth $L$. We have the following lemma.

**Lemma E.5.** *Consider a ReLU network* $f^\infty$ *defined in Eq.(1) with infinite width. For all* $l \in [L]$, *the following relations hold:*

- *for any input* $\mathbf{x} \in \mathbb{R}^d$, $\|\alpha^{(l)}(\mathbf{x})\| = \|\mathbf{x}\|$;

- *for any two inputs* $\mathbf{x}, \mathbf{z} \in \mathbb{R}^d$, $\theta^{(l)}(\mathbf{x}, \mathbf{z}) = g\left(\theta^{(l-1)}(\mathbf{x}, \mathbf{z})\right)$. *Let $g^l(\cdot)$ be the l-fold composition of $g(\cdot)$, then*

$$\theta^{(l)}(\mathbf{x}, \mathbf{z}) = g^{\circ l}\left(\theta_{in}(\mathbf{x}, \mathbf{z})\right). \tag{24}$$

We consider the inner product $\langle \nabla_l f^\infty(\mathbf{z}), \nabla_l f^\infty(\mathbf{x}) \rangle$, for $l \in [L+1]$.[1] By Eq.(22), we have

$$\langle \nabla_l f^\infty(\mathbf{z}), \nabla_l f^\infty(\mathbf{x}) \rangle = \langle \delta^{(l)}(\mathbf{z}), \delta^{(l)}(\mathbf{x}) \rangle \cdot \langle \alpha^{(l-1)}(\mathbf{z}), \alpha^{(l-1)}(\mathbf{x}) \rangle. \tag{25}$$

For $\langle \alpha^{(l-1)}(\mathbf{z}), \alpha^{(l-1)}(\mathbf{x}) \rangle$, applying Lemma E.5, we have

$$\langle \alpha^{(l-1)}(\mathbf{z}), \alpha^{(l-1)}(\mathbf{x}) \rangle = \|\mathbf{x}\| \|\mathbf{z}\| \cos \theta^{(l-1)}(\mathbf{x}, \mathbf{z}). \tag{26}$$

For $\langle \delta^{(l)}(\mathbf{z}), \delta^{(l)}(\mathbf{x}) \rangle$, by definition Eq.(23), we have

$$\langle \delta^{(l)}(\mathbf{z}), \delta^{(l)}(\mathbf{x}) \rangle = \left(\frac{2}{m}\right)^{L-l+1}$$
$$\times W^{(L+1)} \mathbb{I}_{\{\tilde{\alpha}^{(L)}(\mathbf{x}) \geq 0\}} \cdots \underbrace{W^{(l+1)} \mathbb{I}_{\{\tilde{\alpha}^{(l)}(\mathbf{x}) \geq 0, \tilde{\alpha}^{(l)}(\mathbf{z}) \geq 0\}} W^{(l+1)T}}_{A} \cdots \mathbb{I}_{\{\tilde{\alpha}^{(L)}(\mathbf{z}) \geq 0\}} W^{(L+1)T}$$

Recalling that $\tilde{\alpha}^{(l)} = W^{(l)} \tilde{\alpha}^{(l-1)}$ and applying Lemma E.3 on the the term $A$ above, we obtain

$$\langle \delta^{(l)}(\mathbf{z}), \delta^{(l)}(\mathbf{x}) \rangle = \frac{\pi - \theta^{(l-1)}(\mathbf{x}, \mathbf{z})}{\pi} \langle \delta^{(l+1)}(\mathbf{z}), \delta^{(l+1)}(\mathbf{x}) \rangle.$$

Recursively applying the above formula for $l' = l, l+1, \cdots, L$, and noticing that $\delta^{(L+1)} = 1$, we have

$$\langle \delta^{(l)}(\mathbf{z}), \delta^{(l)}(\mathbf{x}) \rangle = \prod_{l'=l-1}^{L-1} \left(1 - \frac{\theta^{(l')}(\mathbf{x}, \mathbf{z})}{\pi}\right). \tag{27}$$

Combining Eq.(25), (26) and (27), we have

$$\langle \nabla_l f^\infty(\mathbf{z}), \nabla_l f^\infty(\mathbf{x}) \rangle = \|\mathbf{x}\| \|\mathbf{z}\| \cos \theta^{(l-1)}(\mathbf{x}, \mathbf{z}) \prod_{l'=l-1}^{L-1} \left(1 - \frac{\theta^{(l')}(\mathbf{x}, \mathbf{z})}{\pi}\right). \tag{28}$$

For the inner product between the full model gradients, we have

$$\langle \nabla f^\infty(\mathbf{z}), \nabla f^\infty(\mathbf{x}) \rangle = \sum_{l=1}^{L+1} \langle \nabla_l f^\infty(\mathbf{z}), \nabla_l f^\infty(\mathbf{x}) \rangle = \|\mathbf{x}\| \|\mathbf{z}\| \sum_{l=0}^{L} \left[\cos \theta^{(l)}(\mathbf{x}, \mathbf{z}) \prod_{l'=l}^{L-1} \left(1 - \frac{\theta^{(l')}(\mathbf{x}, \mathbf{z})}{\pi}\right)\right]. \tag{29}$$

Putting $\mathbf{x} = \mathbf{z}$ in the above equation, we have $\theta^{(l)}(\mathbf{x}, \mathbf{z}) = 0$ for all $l \in [L]$, and obtain

$$\|\nabla f^\infty(\mathbf{x})\|^2 = \|\mathbf{x}\|^2 \cdot (L+1). \tag{30}$$

Hence, we have, for an infinitely wide neural network,

$$\cos \phi^\infty(\mathbf{x}, \mathbf{z}) = \frac{\langle \nabla f^\infty(\mathbf{z}), \nabla f^\infty(\mathbf{x}) \rangle}{\|\nabla f^\infty(\mathbf{x})\| \|\nabla f^\infty(\mathbf{z})\|} = \frac{1}{L+1} \sum_{l=0}^{L} \left[\cos \theta^{(l)}(\mathbf{x}, \mathbf{z}) \prod_{l'=l}^{L-1} (1 - \theta^{(l')}(\mathbf{x}, \mathbf{z})/\pi)\right]. \tag{31}$$

Now, we consider the finitely wide neural network $f$. As have been shown by [17, 9, 21],

$$\langle \nabla f(\mathbf{z}), \nabla f(\mathbf{x}) \rangle - \langle \nabla f^\infty(\mathbf{z}), \nabla f^\infty(\mathbf{x}) \rangle = O\left(\frac{1}{\sqrt{m}}\right), \tag{32}$$

with high probability of random initialization of the network $f$. Letting $\mathbf{z} = \mathbf{x}$ above, we also have

$$\|\nabla f(\mathbf{x})\|^2 = \|\nabla f^\infty(\mathbf{x})\|^2 + O\left(\frac{1}{\sqrt{m}}\right). \tag{33}$$

Using the above two equations, we have

$$\cos \phi(\mathbf{x}, \mathbf{z}) = \frac{\langle \nabla f(\mathbf{z}), \nabla f(\mathbf{x}) \rangle}{\|\nabla f(\mathbf{x})\| \|\nabla f(\mathbf{z})\|} = \cos \phi^\infty(\mathbf{x}, \mathbf{z}) + O\left(\frac{1}{\sqrt{m}}\right), \tag{34}$$

with high probability of random initialization of the network $f$. □

---

[1]With a bit of abuse of notation, we refer to the flattened vectors of $\nabla_l f$ in the inner product.

## E.3 Proof of Theorem 3.2

*Proof.* For simplicity of notation, we don't explicitly write out the dependent on the inputs $\mathbf{x}, \mathbf{z}$, and write $\theta^{(l)} \triangleq \theta^{(l)}(\mathbf{x}, \mathbf{z})$, and $\phi \triangleq \phi(\mathbf{x}, \mathbf{z})$. We start the proof with the summation term on the R.H.S. of Eq. 7 in Lemma 3.1.

$$\frac{1}{L+1} \sum_{l=0}^{L} \left[ \cos \theta^{(l)} \prod_{l'=l}^{L-1} (1 - \theta^{(l')}/\pi) \right]$$

$$\overset{(a)}{=} \frac{1}{L+1} \sum_{l=0}^{L} \left[ \cos \theta^{(0)} \prod_{l'=0}^{l-1} \left( 1 + \frac{1}{\pi} \tan \theta^{(l')} - \frac{1}{\pi} \theta^{(l')} \right) \prod_{l'=l}^{L-1} (1 - \theta^{(l')}/\pi) \right]$$

$$\overset{(b)}{=} \frac{1}{L+1} \sum_{l=0}^{L} \left[ \cos \theta^{(0)} \prod_{l'=0}^{l-1} \left( 1 + \frac{1}{3\pi} (\theta^{(l')})^3 + o(\theta^{(l')})^3 \right) \prod_{l'=l}^{L-1} (1 - \theta^{(l')}/\pi) \right]$$

$$\overset{(c)}{=} \frac{\cos \theta^{(0)}}{L+1} \sum_{l=0}^{L} \left[ \prod_{l'=0}^{l-1} \left( 1 + \frac{1}{3\pi} (\theta^{(0)})^3 + o(\theta^{(0)})^3 \right) \right.$$

$$\left. \times \prod_{l'=l}^{L-1} \left( 1 - \frac{1}{\pi} \theta^{(0)} + \frac{l'}{3\pi^2} (\theta^{(0)})^2 + o((\theta^{(0)})^2) \right) \right]$$

$$= \frac{\cos \theta^{(0)}}{L+1} \sum_{l=0}^{L} \left( 1 - \frac{L-l}{\pi} \theta^{(0)} + \frac{(L-l)(2L-l-2)}{3\pi^2} (\theta^{(0)})^2 + o((\theta^{(0)})^2) \right)$$

$$= \cos \theta^{(0)} \left( 1 - \frac{L}{2\pi} \theta^{(0)} + o(\theta^{(0)}) \right).$$

In step (a) above, we use the relation $\theta^{(l)} = g(\theta^{(l-1)})$, i.e., $\cos \theta^{(l)} = 1 - \cos \theta^{(l-1)}/\pi + \sin \theta^{(l-1)}/\pi$; in step (b), we used the fact that $\theta^{(l)} < \theta_{in}$ (Theorem A.2) which stays small and used the Taylor expansion of $tan$. In step (c), we used the following lemma (proof is in Appendix G.7):

**Lemma E.6.** *Given any inputs $\mathbf{x}, \mathbf{z}$ such that $\theta_{in}(\mathbf{x}, \mathbf{z}) \ll 1$, for each $l \in [L]$, the $l$-embedding angle $\theta^{(l)}(\mathbf{x}, \mathbf{z})$ can be expressed as*

$$\theta^{(l)}(\mathbf{x}, \mathbf{z}) = \theta_{in}(\mathbf{x}, \mathbf{z}) - \frac{l}{3\pi} (\theta_{in}(\mathbf{x}, \mathbf{z}))^2 + o\left( (\theta_{in}(\mathbf{x}, \mathbf{z}))^2 \right).$$

By Lemma 3.1, there exists a constant $c$ such that

$$\cos \phi(\mathbf{x}, \mathbf{z}) < \left( 1 - \frac{L}{2\pi} \theta_{in} + o(\theta_{in}) \right) \cos \theta_{in} + \frac{c}{\sqrt{m}}. \tag{35}$$

When $m > \frac{16\pi^2 c^2}{L^2 \theta_{in}^2 \cos^2 \theta_{in}}$, we have $\cos \phi(\mathbf{x}, \mathbf{z}) < \cos \theta_{in}(\mathbf{x}, \mathbf{z})$, namely, $\phi(\mathbf{x}, \mathbf{z}) > \theta_{in}(\mathbf{x}, \mathbf{z})$. $\qquad\square$

## E.4 Proof of Theorem 3.5

*Proof.* We start the proof with an analysis of the embedding angles $\theta^{(l)}$ in the infinite depth limit.

First, by Lemma E.5 and Proposition A.1, we easily find that, for all input angle $\theta_{in} = \theta^{(0)} \neq 0$, $\theta^{(l)}$ is monotonically decreasing and converges to zero: $\lim_{l \to \infty} \theta^{(l)} \to 0$. As the following analysis is independent of $\theta^{(0)}$, we will not explicitly write out the arguments $\mathbf{x}$ and $\mathbf{z}$.

Now, we analyze its convergence rate, utilizing Eq.(57). As we are considering the infinite depth limit and $\theta^{(l)}$ converges to zero, we can drop its $o(\cdot)$ term and rewrite Eq.(57) as:

$$\frac{d\theta^{(l)}}{dl} = -\frac{1}{3\pi} (\theta^{(l)})^2. \tag{36}$$

Solving this differential equation, we get $\theta^{(l)} = \theta^{(0)}/(1 + (3\pi)^{-1} \theta^{(0)} l)$, and

$$\lim_{l \to \infty} \theta^{(l)} \cdot l = 3\pi. \tag{37}$$

By Eq.(7), we get the following relation between $\phi_L$ and $\phi_{L+1}$:

$$(L+2)\cos\phi_{L+1} = (1 - \theta^{(L)}/\pi) \cdot (L+1)\cos\phi_L + \cos\theta^{(L+1)}. \tag{38}$$

Rearranging terms, we get

$$(\cos\phi_{L+1} - \cos\phi_L) = -\left(1 + \frac{(L+1)\theta^{(L)}}{\pi}\right)\frac{\cos\phi_L}{L+2} + \frac{1}{L+2}\cos\theta^{(L+1)}. \tag{39}$$

The right side of Eq.(39) converges to zero as $L \to \infty$. Using Eq.(37) and $\lim_{L\to\infty}\cos\theta^{(L+1)} = 1$, we have

$$\lim_{L\to\infty}\cos\phi_L = \frac{1}{4}. \tag{40}$$

Hence, $\phi_L$ converges to $\arccos\frac{1}{4} \approx 75.5°$, in the limit $L \to \infty$.

$\square$

### E.5 Proof of Theorem 4.2

*Proof.* First of all, we note that in scenario (b), i.e., the network with all ReLU activation removed, the network simply becomes a linear neural network (while with the same trainable parameters $W^{(1)}$ as the ReLU network in scenario (a)). By the analysis in Section 2.1, we can easily see that the NTK matrix in scenario (b) is equivalent to the Gram matrix $G$, and $\kappa_b = \kappa_0$. Hence, whenever comparing the two scenarios, it suffices to compare the NTK $K$ (and its condition number $\kappa$) of ReLU network with the Gram matrix $G$ (and its condition number $\kappa_0$).

We prove the theorem by induction.

**Base case: ReLU neural network of depth $L = 1$.** First, consider the shallow ReLU neural network

$$f(W; \mathbf{x}) = \frac{\sqrt{2}}{\sqrt{m}}\mathbf{v}^T\sigma(W\mathbf{x}), \tag{41}$$

where $W$ are the trainable parameters.

The model gradient, for an arbitrary input $\mathbf{x}$, can be written as

$$\nabla f(\mathbf{x}) = \mathbf{x}\delta(\mathbf{x}) \in \mathbb{R}^{d\times m}, \tag{42}$$

where $\delta(\mathbf{x}) \in \mathbb{R}^{1\times m}$ has the following expression

$$\delta(\mathbf{x}) = \sqrt{\frac{2}{m}}\mathbf{v}^T\mathbb{I}_{\{W\mathbf{x}\geq 0\}}.$$

At initialization, $W$ is a random matrix. Recall that the NTK $K = FF^T$, where the gradient feature matrix $F$ consist of the gradient feature vectors $\nabla f(\mathbf{x})$ for all $\mathbf{x}$ for the dataset. Applying Lemma D.1 in the limit of $m \to \infty$, we have that each entry $K_{ij}$ is equivalent to $\sum_{k,l} \mathbf{A}'_{ikl}\mathbf{A}'_{jkl}$, with $\mathbf{A}'_{ikl} := \sqrt{2}X_{ik}\mathbb{I}_{\{W_{l:}X_{i:}\geq 0\}}$, where $X \in \mathbb{R}^{n\times d}$ is the matrix of input data. Then apply Lemma E.4, we immediately have that

$$\lambda_{min}(K) > \lambda_{min}(G), \quad \lambda_{max}(K) < \lambda_{max}(G).$$

Hence, we have that $\kappa_a < \kappa_b$.

In addition, note that this network has one hidden layer, and that the "zero-hidden layer" network is just simply the linear model. For linear model, the NTK is simply the Gram matrix. Hence, for the base case, we have $\kappa_{f_1} < \kappa_{f_2} = \kappa_0$, with network $f_1$ of depth $1$ and network $f_2$ of depth $0$.

**Induction hypothesis.** Suppose that, for a ReLU network $f_{L-1}$ of depth $L - 1$, its NTK condition number $\kappa_{L-1}$ is strictly smaller than $\kappa_0$.

**Induction step.** Now, let's consider the two ReLU networks $f_L$ of depth $L$ and $f_{L-1}$. It is suffices to prove that $\kappa_L < \kappa_{L-1}$. The model gradients, for any given input $\mathbf{x}$, can be written as:

$$\nabla f_L(\mathbf{x}) = \mathbf{x}\delta_L(\mathbf{x}) \in \mathbb{R}^{d \times m}, \quad \nabla f_{L-1}(\mathbf{x}) = \mathbf{x}\delta_{L-1}(\mathbf{x}) \in \mathbb{R}^{d \times m},$$

where

$$\delta_L(\mathbf{x}) = \sqrt{\frac{2}{m}}W^{(L+1)}\mathbb{I}_{\{W^{(L)}\alpha^{(L-1)} \geq 0\}}\sqrt{\frac{2}{m}}W^{(L)}\mathbb{I}_{\{W^{(L-1)}\alpha^{(L-2)} \geq 0\}} \cdots \sqrt{\frac{2}{m}}W^{(2)}\mathbb{I}_{\{W^{(1)}\alpha^{(0)} \geq 0\}}$$

$$\delta_{L-1}(\mathbf{x}) = \sqrt{\frac{2}{m}}W^{(L)}\mathbb{I}_{\{W^{(L-1)}\alpha^{(L-2)} \geq 0\}} \cdots \sqrt{\frac{2}{m}}W^{(2)}\mathbb{I}_{\{W^{(1)}\alpha^{(0)} \geq 0\}}$$

Note that the matrix $W^{(L)}$ has different dimensions for $f_L$ and $f_{L-1}$.

Using the same argument as in the base case, as well as applying Lemma D.1 when contracting the $\delta(\mathbf{x})$'s, we directly obtain $\kappa_L < \kappa_{L-1}$. $\qquad\square$

### E.6 Proof of Theorem 4.3

*Proof.* First, consider the normalized NTK matrix $\frac{1}{L+1}K$. By Lemma 3.1, we have for it diagonal elements:

$$\frac{1}{L+1}K(\mathbf{x}_i, \mathbf{x}_i) = \frac{1}{L+1}\|\nabla f(\mathbf{x})\|^2 = \|\mathbf{x}_i\|^2 = 1. \tag{43}$$

By Theorem 3.5, we have that, in the infinite depth limit, each of off-diagonal elements of the normalized NTK matrix converges to $\frac{1}{4}$. Namely,

$$\frac{1}{L+1}K \to \begin{bmatrix} 1 & \frac{1}{4} & \frac{1}{4} & \cdots & \frac{1}{4} \\ \frac{1}{4} & 1 & \frac{1}{4} & \cdots & \frac{1}{4} \\ \frac{1}{4} & \frac{1}{4} & 1 & \cdots & \frac{1}{4} \\ \vdots & \vdots & \vdots & \ddots & \vdots \\ \frac{1}{4} & \frac{1}{4} & \frac{1}{4} & \cdots & 1 \end{bmatrix} = \frac{3}{4}I_n + \frac{1}{4}J_n, \tag{44}$$

where matrix $J_n$ has its all elements being ones. Therefore, $\lim_{L\to\infty}\frac{1}{L+1}K$ has one eigenvalue $\lambda_1 = 1 + \frac{n}{4}$, and all remaining eigenvalues $\lambda_2 = \lambda_3 = \cdots = \lambda_n = \frac{3}{4}$. Then its condition number is $\kappa = \frac{\lambda_1}{\lambda_n} = \frac{4+n}{3}$. $\qquad\square$

## F  Relaxing the constraint on top layers

**Theorem F.1.** *Consider a $L$-layer ReLU neural network $f$ as defined in Eq.(1) in the infinite width limit $m \to \infty$ and at initialization. We compare the NTK condition numbers $\kappa_a$ and $\kappa_b$ of the two scenarios: (a) the network with the ReLU activation, and (b) the network with all the ReLU activation removed. Consider the dataset $\mathcal{D} = \{(\mathbf{x}_1, y_1), (\mathbf{x}_2, y_2)\}$ with the input angle $\theta_{in}$ between $\mathbf{x}_1$ and $\mathbf{x}_2$ small, $\theta_{in} \ll 1$. Then, the NTK condition number $\kappa_a < \kappa_b$. Moreover, for two ReLU neural networks $f_1$ of depth $L_1$ and $f_2$ of depth $L_2$ with $L_1 > L_2$, we have $\kappa_{f_1} < \kappa_{f_2}$.*

*Proof.* First, let's consider the scenario (a), i.e. the ReLU network. According to the definition of NTK and Lemma 3.1, the NTK matrix $K$ for this dataset $\mathcal{D} = \{(\mathbf{x}_1, y_1), (\mathbf{x}_2, y_2)\}$ is (NTK is normalized by the factor $1/(L+1)^2$):

$$K = \begin{pmatrix} \|\nabla f(\mathbf{x}_1)\|^2 & \langle \nabla f(\mathbf{x}_1), \nabla f(\mathbf{x}_2)\rangle \\ \langle \nabla f(\mathbf{x}_2), \nabla f(\mathbf{x}_1)\rangle & \|\nabla f(\mathbf{x}_2)\|^2 \end{pmatrix} = \begin{pmatrix} \|\mathbf{x}_1\|^2 & \|\mathbf{x}_1\|\|\mathbf{x}_2\|\cos\phi \\ \|\mathbf{x}_1\|\|\mathbf{x}_2\|\cos\phi & \|\mathbf{x}_2\|^2 \end{pmatrix}.$$

The eigenvalues of the NTK matrix $K$ are given by

$$\lambda_1(K) = \frac{1}{2}\left(\|\mathbf{x}_1\|^2 + \|\mathbf{x}_2\|^2 + \sqrt{\|\mathbf{x}_1\|^4 + \|\mathbf{x}_2\|^4 + \|\mathbf{x}_1\|^2\|\mathbf{x}_2\|^2\cos 2\phi}\right), \tag{45a}$$

$$\lambda_2(K) = \frac{1}{2}\left(\|\mathbf{x}_1\|^2 + \|\mathbf{x}_2\|^2 - \sqrt{\|\mathbf{x}_1\|^4 + \|\mathbf{x}_2\|^4 + \|\mathbf{x}_1\|^2\|\mathbf{x}_2\|^2\cos 2\phi}\right). \tag{45b}$$

In the scenario (b), the ReLU activation is removed in the network, resulting in a linear neural network. In this case, the NTK is equivalent to the Gram matrix $G$, as given by Corollary 2.4. We have

$$G = \begin{pmatrix} \|\mathbf{x}_1\|^2 & \mathbf{x}_1^T\mathbf{x}_2 \\ \mathbf{x}_1^T\mathbf{x}_2 & \|\mathbf{x}_2\|^2 \end{pmatrix} = \begin{pmatrix} \|\mathbf{x}_1\|^2 & \|\mathbf{x}_1\|\|\mathbf{x}_2\|\cos\theta_{in} \\ \|\mathbf{x}_1\|\|\mathbf{x}_2\|\cos\theta_{in} & \|\mathbf{x}_2\|^2 \end{pmatrix},$$

and its eigenvalues as

$$\lambda_1(G) = \frac{1}{2}\left(\|\mathbf{x}_1\|^2 + \|\mathbf{x}_2\|^2 + \sqrt{\|\mathbf{x}_1\|^4 + \|\mathbf{x}_2\|^4 + \|\mathbf{x}_1\|^2\|\mathbf{x}_2\|^2\cos 2\theta_{in}}\right),$$

$$\lambda_2(G) = \frac{1}{2}\left(\|\mathbf{x}_1\|^2 + \|\mathbf{x}_2\|^2 - \sqrt{\|\mathbf{x}_1\|^4 + \|\mathbf{x}_2\|^4 + \|\mathbf{x}_1\|^2\|\mathbf{x}_2\|^2\cos 2\theta_{in}}\right).$$

By Theorem 3.2, we have $\cos\phi < \cos\theta_{in}$, when $\theta_{in} \ll 1$ and $\theta_{in} \neq 0$. Hence, we have the following relations

$$\lambda_1(G) > \lambda_1(K) > \lambda_2(K) > \lambda_2(G),$$

which immediately implies $\kappa_a < \kappa_b$.

When comparing ReLU networks with different depths, i.e., network $f_1$ with depth $L_1$ and network $f_2$ with depth $L_2$ with $L_1 > L_2$, notice that in Eq.(45) the top eigenvalue $\lambda_1$ monotonically decreases in $\phi$, and the bottom (smaller) eigenvalue $\lambda_2$ monotonically increases in $\phi$. By the proof of Theorem 3.2, we know that the deeper ReLU network $f_1$ has a better separation than the shallower one $f_2$, i.e., $\phi_{f_1} > \phi_{f_2}$. Hence, we get

$$\lambda_1(K_{f_2}) > \lambda_1(K_{f_1}) > \lambda_2(K_{f_1}) > \lambda_2(K_{f_2}). \tag{46}$$

Therefore, we obtain $\kappa_{f_1} < \kappa_{f_2}$. Namely the deeper ReLU network has a smaller NTK condition number. $\qquad\square$

## G   Technical proofs

### G.1   Proof of Lemma D.1

*Proof.* We denote $A_{ij}$ as the $(i,j)$-th entry of the matrix $A$. Therefore, $(A^T A)_{ij} = \sum_{k=1}^m A_{ki}A_{kj}$. First we find the mean of each $(A^T A)_{ij}$. Since $A_{ij}$ are i.i.d. and has zero mean, we can easily see that for any index $k$,

$$\mathbb{E}[A_{ki}A_{kj}] = \begin{cases} 1, & \text{if } i = j \\ 0, & \text{otherwise} \end{cases}.$$

Consequently,

$$\mathbb{E}[(\frac{1}{m}A^T A)_{ij}] = \begin{cases} 1, & \text{if } i = j \\ 0, & \text{otherwise} \end{cases}.$$

That is $\mathbb{E}[\frac{1}{m}A^T A] = I_d$.

Now we consider the variance of each $(A^T A)_{ij}$. If $i \neq j$ we can explicitly write,

$$Var\left[\frac{1}{m}(A^T A)_{ij}\right] = \frac{1}{m^2} \cdot \mathbb{E}\left[\sum_{k_1=1}^m \sum_{k_2=1}^m A_{k_1 i}A_{k_1 j}A_{k_2 i}A_{k_2 j}\right]$$

$$= \frac{1}{m^2} \cdot \sum_{k_1=1}^m \sum_{k_2=1}^m \mathbb{E}\left[A_{k_1 i}A_{k_1 j}A_{k_2 i}A_{k_2 j}\right]$$

$$= \frac{1}{m^2}\left(\sum_{k=1}^m \mathbb{E}\left[A_{ki}^2 A_{kj}^2\right] + \sum_{k_1 \neq k_2} \mathbb{E}\left[A_{k_1 i}A_{k_1 j}A_{k_2 i}A_{k_2 j}\right]\right)$$

$$= \frac{1}{m^2}\left(\sum_{k=1}^m \mathbb{E}\left[A_{ki}^2\right]\mathbb{E}\left[A_{kj}^2\right] + \sum_{k_1 \neq k_2} \mathbb{E}[A_{k_1 i}]\mathbb{E}[A_{k_1 j}]\mathbb{E}[A_{k_2 i}]\mathbb{E}[A_{k_2 j}]\right)$$

$$= \frac{1}{m^2} \cdot (m + 0) = \frac{1}{m}.$$

In the case of $i = j$, then,

$$Var\left[\frac{1}{m}(A^T A)_{ii}\right] = \frac{1}{m^2} \cdot Var\left[\sum_{k=1}^{m} A_{ki}^2\right] = \frac{1}{m^2} \cdot \sum_{k=1}^{m} Var\left[A_{ki}^2\right] \overset{(a)}{=} \frac{1}{m^2}(m \cdot 2) = \frac{2}{m}. \quad (47)$$

In the equality (a) above, we used the fact that $A_{ki}^2 \sim \chi^2(1)$. Therefore, $\lim_{m \to \infty} Var(\frac{1}{m}(A^T A)) = 0$.

Now applying Chebyshev's inequality we get,

$$Pr(|\frac{1}{m}A^T A - I_d| \geq \epsilon) \leq \frac{Var(\frac{1}{m}(A^T A))}{\epsilon} \quad (48)$$

Obviously for any $\epsilon \geq 0$ as $m \to \infty$, the R.H.S. goes to zero. Thus, $\frac{1}{m}A^T A \to I_{d \times d}$, in probability.
□

## G.2 Proof of Lemma E.1

*Proof.* Note that the random vector $\mathbf{w}$ is isotropically distributed and that only inner products $\mathbf{w}^T \mathbf{v}_1$ and $\mathbf{w}^T \mathbf{v}_2$ appear, hence we can assume without loss of generality that (if not, one can rotate the coordinate system to make it true):

$$\mathbf{v}_1 = \|\mathbf{v}_1\|(1, 0, 0, \cdots, 0),$$
$$\mathbf{v}_2 = \|\mathbf{v}_2\|(\cos\theta, \sin\theta, 0, \cdots, 0).$$

In this setting, the only relevant parts of $\mathbf{w}$ are its first two scalar components $w_1$ and $w_2$. Define $\tilde{\mathbf{w}}$ as

$$\tilde{\mathbf{w}} = (w_1, w_2, 0, \cdots, 0) = \sqrt{w_1^2 + w_2^2}(\cos\omega, \sin\omega, 0, \cdots, 0). \quad (49)$$

Then,

$$\mathbb{P}[(\mathbf{w}^T \mathbf{v}_1 \geq 0) \wedge (\mathbf{w}^T \mathbf{v}_2 \geq 0)] = \mathbb{P}[(\tilde{\mathbf{w}}^T \mathbf{v}_1 \geq 0) \wedge (\tilde{\mathbf{w}}^T \mathbf{v}_2 \geq 0)] = \frac{1}{2\pi} \int_{\theta - \frac{\pi}{2}}^{\frac{\pi}{2}} d\omega = \frac{1}{2} - \frac{\theta}{2\pi}.$$

□

## G.3 Proof of Lemma E.2

*Proof.* Note that the ReLU activation function $\sigma(z)$ can be written as $z\mathbb{I}_{z \geq 0}$. We have,

$$\langle \mathbf{u}_1, \mathbf{u}_2 \rangle = \frac{2}{q} \mathbf{v}_1^T W^T \mathbb{I}_{\{W\mathbf{v}_1 \geq 0, W\mathbf{v}_2 \geq 0\}} W \mathbf{v}_2$$

$$= \frac{2}{q} \sum_{i=1}^{q} \mathbf{v}_1^T (W_{i \cdot})^T \mathbb{I}_{\{W_{i \cdot} \mathbf{v}_1 \geq 0, W_{i \cdot} \mathbf{v}_2 \geq 0\}} W_{i \cdot} \mathbf{v}_2$$

$$\overset{q \to \infty}{=} 2\mathbb{E}_{\mathbf{w} \sim \mathcal{N}(0, I_{p \times p})}[\mathbf{v}_1^T \mathbf{w} \mathbb{I}_{\{\mathbf{w}^T \mathbf{v}_1 \geq 0, \mathbf{w}^T \mathbf{v}_2 \geq 0\}} \mathbf{w}^T \mathbf{v}_2]$$

Note that the random vector $\mathbf{w}$ is isotropically distributed and that only inner products $\mathbf{w}^T \mathbf{v}_1$ and $\mathbf{w}^T \mathbf{v}_2$ appear, hence we can assume without loss of generality that (if not, one can rotate the coordinate system to make it true):

$$\mathbf{v}_1 = \|\mathbf{v}_1\|(1, 0, 0, \cdots, 0),$$
$$\mathbf{v}_2 = \|\mathbf{v}_2\|(\cos\theta, \sin\theta, 0, \cdots, 0).$$

In this setting, the only relevant parts of $\mathbf{w}$ are its first two scalar components $w_1$ and $w_2$. Define $\tilde{\mathbf{w}}$ as

$$\tilde{\mathbf{w}} = (w_1, w_2, 0, \cdots, 0) = \sqrt{w_1^2 + w_2^2}(\cos\omega, \sin\omega, 0, \cdots, 0). \quad (50)$$

Then, in the limit of $q \to \infty$,

$$
\begin{aligned}
\langle \mathbf{u}_1, \mathbf{u}_2 \rangle &= 2\mathbb{E}_{\mathbf{w} \sim \mathcal{N}(0, I_{p \times p})}[\mathbf{v}_1^T \mathbf{w} \mathbb{I}_{\{\mathbf{w}^T \mathbf{v}_1 \geq 0, \mathbf{w}^T \mathbf{v}_2 \geq 0\}} \mathbf{w}^T \mathbf{v}_2] \\
&= 2\mathbb{E}_{\tilde{\mathbf{w}} \sim \mathcal{N}(0, I_{2 \times 2})}[\mathbf{v}_1^T \tilde{\mathbf{w}} \mathbb{I}_{\{\tilde{\mathbf{w}}^T \mathbf{v}_1 \geq 0, \tilde{\mathbf{w}}^T \mathbf{v}_2 \geq 0\}} \tilde{\mathbf{w}}^T \mathbf{v}_2] \\
&= 2\|\mathbf{v}_1\| \|\mathbf{v}_2\| \cdot \mathbb{E}_{\tilde{\mathbf{w}} \sim \mathcal{N}(0, I_{2 \times 2})}[\|\tilde{\mathbf{w}}\|^2] \cdot \frac{1}{2\pi} \int_{\theta - \frac{\pi}{2}}^{\frac{\pi}{2}} \cos \omega \cos(\theta - \omega) d\omega \\
&= 2\|\mathbf{v}_1\| \|\mathbf{v}_2\| \cdot 2 \cdot \frac{1}{4\pi} \left( (\pi - \theta) \cos \theta + \sin \theta \right) \\
&= \|\mathbf{v}_1\| \|\mathbf{v}_2\| \frac{1}{\pi} \left( (\pi - \theta) \cos \theta + \sin \theta \right).
\end{aligned}
$$

$\square$

## G.4 Proof of Lemma E.3

*Proof.*

$$
\begin{aligned}
A_1 A_2^T &= \frac{2}{q} \sum_{k=1}^{q} U_{\cdot k} \mathbb{I}_{\{W_{k \cdot} \mathbf{v}_1 \geq 0, W_{k \cdot} \mathbf{v}_2 \geq 0\}} (U_{\cdot k})^T \\
&\stackrel{q \to \infty}{=} 2 \cdot \mathbb{E}_{\mathbf{u} \sim \mathcal{N}(0, I_{s \times s}), \mathbf{w} \sim \mathcal{N}(0, I_{p \times p})}[\mathbf{u} \mathbf{u}^T \mathbb{I}_{\{\mathbf{w}^T \mathbf{v}_1 \geq 0, \mathbf{w}^T \mathbf{v}_2 \geq 0\}}] \\
&\stackrel{(a)}{=} 2 \cdot \mathbb{E}_{\mathbf{u} \sim \mathcal{N}(0, I_{s \times s})}[\mathbf{u} \mathbf{u}^T] \cdot \mathbb{E}_{\mathbf{w} \sim \mathcal{N}(0, I_{p \times p})}[\mathbb{I}_{\{\mathbf{w}^T \mathbf{v}_1 \geq 0, \mathbf{w}^T \mathbf{v}_2 \geq 0\}}] \\
&= 2 \cdot \mathbb{E}_{\mathbf{u} \sim \mathcal{N}(0, I_{s \times s})}[\mathbf{u} \mathbf{u}^T] \cdot \mathbb{P}[(\mathbf{w}^T \mathbf{v}_1 \geq 0) \wedge (\mathbf{w}^T \mathbf{v}_2 \geq 0)] \\
&\stackrel{(b)}{=} \frac{\pi - \theta}{\pi} I_{s \times s}.
\end{aligned}
$$

In the step $(a)$ above, we used the fact that $U$ is independent of $W$, $\mathbf{v}_1$ and $\mathbf{v}_2$. In the step $(b)$ above, we applied Lemma E.1, and used the fact that $\mathbb{E}_{\mathbf{u} \sim \mathcal{N}(0, I_{s \times s})}[\mathbf{u} \mathbf{u}^T] = I_{s \times s}$. $\square$

## G.5 Proof of Lemma E.4

*Proof.* Starting from the definition of the smallest eigenvalue, we have that $\lambda_{min}(B')$ satisfies

$$
\begin{aligned}
\lambda_{min}(B') &= \min_{\mathbf{u} \neq 0} \frac{\mathbf{u}^T B' \mathbf{u}}{\|\mathbf{u}\|^2} \\
&= \min_{\mathbf{u} \neq 0} \frac{\sum_{l=1}^{q} \sum_{k=1}^{p} (\sum_{i=1}^{n} \sqrt{2} u_i A_{ik} \mathbb{I}_{\{W_{l:} A_{i:} \geq 0\}})^2}{\sum_{i=1}^{n} u_i^2} \\
&= \min_{\mathbf{u} \neq 0} \sum_{l=1}^{q} \frac{\sum_{i=1}^{n} 2(u_i \mathbb{I}_{\{W_{l:} A_{i:} \geq 0\}})^2}{\sum_{i=1}^{n} u_i^2} \frac{\sum_{k=1}^{p} (\sum_{i=1}^{n} \sqrt{2} u_i A_{ik} \mathbb{I}_{\{W_{l:} A_{i:} \geq 0\}})^2}{\sum_{i=1}^{n} 2(u_i \mathbb{I}_{\{W_{l:} A_{i:} \geq 0\}})^2} \\
&\stackrel{(a)}{>} \min_{\mathbf{u} \neq 0} \sum_{l=1}^{q} \frac{\sum_{i=1}^{n} 2(u_i \mathbb{I}_{\{W_{l:} A_{i:} \geq 0\}})^2}{\sum_{i=1}^{n} u_i^2} \lambda_{min}(B). \quad (51)
\end{aligned}
$$

In the inequality $(a)$ above, we made the following treatment: for each fixed $l$, we consider $u_i \mathbb{I}_{\{W_{l:} A_{i:} \geq 0\}}$ as the $i$-th component of a vector $\mathbf{u}_l'$; by definition, the minimum eigenvalue of matrix $B = AA^T$

$$
\lambda_{min}(B) = \min_{\mathbf{u}' \neq 0} (\mathbf{u}')^T B \mathbf{u}' / \|\mathbf{u}'\|^2 \leq (\mathbf{u}_j')^T B \mathbf{u}_j' / \|\mathbf{u}_j'\|^2, \ \forall j; \quad (52)
$$

moreover, this $\leq$ inequality becomes equality, if and only if all $\mathbf{u}_j'$ are the same and equal to $\arg\min_{\mathbf{u}' \neq 0} (\mathbf{u}')^T G \mathbf{u}' / \|\mathbf{u}'\|^2$. It is easy to see, when the dataset is not degenerate, for different $j$, $\mathbf{u}_j'$ are different, hence only the strict inequality $<$ holds in step $(a)$.

Continuing from Eq.(51), we have

$$\lambda_{min}(B') > \min_{\mathbf{u} \neq 0} \sum_{l=1}^{q} \frac{\sum_{i=1}^{n} 2(u_i \mathbb{I}_{\{\{W_{l:} A_{i:} \geq 0\}\}})^2}{\sum_{i=1}^{n} u_i^2} \lambda_{min}(B)$$

$$= \min_{\mathbf{u} \neq 0} \frac{\sum_{i=1}^{n} 2u_i^2 \sum_{l=1}^{q} \mathbb{I}_{\{\{W_{l:} A_{i:} \geq 0\}\}})}{\sum_{i=1}^{n} u_i^2} \lambda_{min}(B)$$

$$= \min_{\mathbf{u} \neq 0} \frac{\sum_{i=1}^{n} u_i^2}{\sum_{i=1}^{n} u_i^2} \lambda_{min}(B) = \lambda_{min}(B).$$

Therefore, we have that $\lambda_{min}(B') > \lambda_{min}(B)$.

As for the largest eigenvalue $\lambda_{max}(B')$, we can apply the same logic above for $\lambda_{min}(K)$ (except replacing the $\min$ operator by $\max$ and have $<$ in step $(a)$) to get $\lambda_{max}(B') < \lambda_{max}(B)$. □

### G.6 Proof of Lemma E.5

*Proof.* Consider an arbitrary layer $l \in [L]$ of the ReLU neural network $f$ at initialization. Given two arbitrary network inputs $\mathbf{x}, \mathbf{z} \in \mathbb{R}^d$, the inputs to the $l$-th layer are $\alpha^{(l-1)}(\mathbf{x}))$ and $\alpha^{(l-1)}(\mathbf{z}))$, respectively.

By definition, we have

$$\alpha^{(l)}(\mathbf{x}) = \sqrt{\frac{2}{m}} \sigma \left( W^{(l)} \alpha^{(l-1)}(\mathbf{x}) \right), \quad \alpha^{(l)}(\mathbf{z}) = \sqrt{\frac{2}{m}} \sigma \left( W^{(l)} \alpha^{(l-1)}(\mathbf{z}) \right), \tag{53}$$

with entries of $W^{(l)}$ being i.i.d. drawn from $\mathcal{N}(0, 1)$. Recall that, by definition, the angle between $\alpha^{(l-1)}(\mathbf{x}))$ and $\alpha^{(l-1)}(\mathbf{z}))$ is $\theta^{(l-1)}(\mathbf{x}, \mathbf{z})$. Applying Lemma E.2, we immediately have the inner product

$$\langle \alpha^{(l)}(\mathbf{z}), \alpha^{(l)}(\mathbf{x}) \rangle = \frac{1}{\pi} \left( (\pi - \theta^{(l-1)}(\mathbf{x}, \mathbf{z})) \cos \theta^{(l-1)}(\mathbf{x}, \mathbf{z}) + \sin \theta^{(l-1)}(\mathbf{x}, \mathbf{z}) \right)$$

$$\times \|\alpha^{(l-1)}(\mathbf{x})\| \|\alpha^{(l-1)}(\mathbf{z})\|. \tag{54}$$

In the special case of $\mathbf{x} = \mathbf{z}$, we have $\theta^{(l-1)}(\mathbf{x}, \mathbf{z}) = 0$, and obtain from the above equation that

$$\|\alpha^{(l)}(\mathbf{x})\|^2 = \|\alpha^{(l-1)}(\mathbf{x})\|^2. \tag{55}$$

Apply Eq.(55) back to Eq.(54), we also get

$$\cos \theta^{(l)}(\mathbf{x}, \mathbf{z}) = \frac{\langle \alpha^{(l)}(\mathbf{z}), \alpha^{(l)}(\mathbf{x}) \rangle}{\|\alpha^{(l)}(\mathbf{x})\| \|\alpha^{(l)}(\mathbf{z})\|} = \frac{1}{\pi} \left( (\pi - \theta^{(l-1)}(\mathbf{x}, \mathbf{z})) \cos \theta^{(l-1)}(\mathbf{x}, \mathbf{z}) + \sin \theta^{(l-1)}(\mathbf{x}, \mathbf{z}) \right)$$
$$\tag{56}$$

That is $\theta^{(l)}(\mathbf{x}, \mathbf{z}) = g(\theta^{(l-1)}(\mathbf{x}, \mathbf{z}))$. Recursively apply this relation, we obtain the desired result. □

### G.7 Proof of Lemma E.6

*Proof.* By Lemma E.5, we have that

$$\cos \theta^{(l)}(\mathbf{x}, \mathbf{z}) = \left( 1 - \frac{\theta^{(l-1)}(\mathbf{x}, \mathbf{z})}{\pi} \right) \cos \theta^{(l-1)}(\mathbf{x}, \mathbf{z}) + \frac{1}{\pi} \sin \theta^{(l-1)}(\mathbf{x}, \mathbf{z})$$

$$= \cos \theta^{(l-1)}(\mathbf{x}, \mathbf{z}) \left( 1 + \frac{1}{\pi} \left( \tan \theta^{(l-1)}(\mathbf{x}, \mathbf{z}) - \theta^{(l-1)}(\mathbf{x}, \mathbf{z}) \right) \right)$$

$$= \cos \theta^{(l-1)}(\mathbf{x}, \mathbf{z}) \left( 1 + \frac{1}{3\pi} (\theta^{(l-1)}(\mathbf{x}, \mathbf{z}))^3 + o \left( (\theta^{(l-1)}(\mathbf{x}, \mathbf{z}))^3 \right) \right).$$

Noting that the Taylor expansion of the $\cos$ function at zero is $\cos z = 1 - \frac{1}{2} z^2 + o(z^3)$, one can easily check that, for all $l \in [L]$,

$$\theta^{(l)}(\mathbf{x}, \mathbf{z}) = \theta^{(l-1)}(\mathbf{x}, \mathbf{z}) - \frac{1}{3\pi} (\theta^{(l-1)}(\mathbf{x}, \mathbf{z}))^2 + o \left( (\theta^{(l-1)}(\mathbf{x}, \mathbf{z}))^2 \right). \tag{57}$$

Note that $\theta^{(l)}(\mathbf{x}, \mathbf{z}) \leq \theta^{(l-1)}(\mathbf{x}, \mathbf{z}) = o(1/L)$. Iteratively apply the above equation, one gets, for all $l \in [L]$, if $\theta^{(0)}(\mathbf{x}, \mathbf{z}) = o(1/L)$,

$$\theta^{(l)}(\mathbf{x}, \mathbf{z}) = \theta^{(0)}(\mathbf{x}, \mathbf{z}) - \frac{l}{3\pi}(\theta^{(0)}(\mathbf{x}, \mathbf{z}))^2 + o\left((\theta^{(0)}(\mathbf{x}, \mathbf{z}))^2\right). \tag{58}$$

$\square$

