# OpenReview forum: "Better NTK Conditioning: A Free Lunch from (ReLU) Nonlinear Activation in Wide Neural Networks"
_NeurIPS.cc/2025/Conference — NeurIPS 2025 poster_

### Official Review · Reviewer_gxGU · 2025-06-29

**Clarity:** 3
**Significance:** 2
**Originality:** 3
**Rating:** 4
**Confidence:** 5

**Summary:**

The authors investigate the advantages of using ReLU activation in the Neural Tangent Kernel (NTK) regime of fully connected neural networks. They demonstrate that ReLU improves data separation in the gradient
feature space, leading to better conditioning of the NTK compared to linear activations. This improvement becomes more pronounced as network depth increases. Their findings are supported by experiments on both synthetic and standard benchmark datasets.

**Questions:**

See above.

**Ethical Concerns:**

["NO or VERY MINOR ethics concerns only"]

**Final Justification:**

While the authors’ response only partially addresses my concerns, I think the paper is worth publishing, and I maintain my original score.

**Limitations:**

1. What are the practical implications of your theoretical findings?
2. Could you clarify how the separation property of the data points relates to their corresponding labels?
3. Figure 2 provides numerical evidence that the separation phenomenon also holds for various non-linear activation functions. However, the behaviour differs across ReLU², GeLU, and Tanh. Can your theoretical framework be extended to accommodate other non-linear activations and account for the observed differences?

**Paper Formatting Concerns:**

None.

**Quality:**

3

**Strengths And Weaknesses:**

Strengths

The authors propose a new direction for investigation, focusing on characterizing the angles between data points in the NTK feature space. Additionally, they examine the spectral properties of the NTK, with particular attention to its condition number. Their theoretical insights are supported by a thorough experimental evaluation, showing agreement between empirical findings and theoretical predictions.

Weaknesses

The main limitation of the paper lies in the narrow scope of the comparison. Specifically, the focus is on ReLU versus linear networks. The theoretical analysis emphasizes that ReLU networks yield greater angle
separation for similar data points in the NTK feature space. However, the impact of non-linear activations in neural networks is already well established, making this result less surprising. A more compelling
direction would have been to explore and compare a broader range of activation functions.

In this context, the setup of the trainable networks is rather restricted and lacks practical applicability. Specifically, in Theorem 2.2 (linear network), the feature space is attributed to the gradient at initialization.
In Theorem 4.2 (non-linear network), only the first layer is trainable, while all subsequent layers remain fixed.

Throughout the manuscript, the intuition underlying the separation property is not clearly expressed. In fact, for similar data points, increased separation may be undesired, especially when the labels are the same. Yet, the manuscript does not explore how this separation behaviour is influenced by the labelling of the data.

The spectral properties of the NTK have been extensively studied in prior work. This manuscript focuses specifically on the condition number of the NTK, which is closely tied to its smallest eigenvalue. However, the comparison is again limited to the simplistic case of linear networks. It remains unclear what the practical implications of this comparison are. How does this insight benefit neural network training or inference?

Moreover, while the manuscript claims that the condition number is small (line 57), it appears to be quite large when applied to larger datasets.

Proposition 4.1 has limited applicability, as it is demonstrated only in the context of small datasets where the number of samples n is less than the dimensionality d.

---

> ### Author Rebuttal · Authors · 2025-07-30
>
> We thank the reviewer for the comment. We address your concerns and questions below.
>
> **Re**: “*The main limitation of the paper lies in the narrow scope of the comparison. Specifically, the focus is on ReLU versus linear networks.… A more compelling direction would have been to explore and compare a broader range of activation functions.*”
>
> **A**: We’d like to point out that our goal is to extract the influence by the existence of non-linear activations. The only way to fulfil this is to directly compare non-linear activated NN with linear NN. Comparing among different non-linear activations does not work for this goal. Analogously, please think about another important property of non-linear activations – increasing expressivity of NNs: with non-linear activations NN can potentially approximate any continuous function; without it is basically a linear model.
>
> In addition, we indeed considered other activation functions, as shown in Figure 2. Again, these need to be compared with linear NNs in order to see the influence of these activations.
>
> **Re**: “*... the impact of non-linear activations in neural networks is already well established, making this result less surprising.*”
>
> **A**: We didn’t find any similar work in the literature, and we don't think our result is previously known. We’d like to ask the reviewer to provide additional clarification and references.
>
> **Re**: “*... in Theorem 2.2 (linear network), the feature space is attributed to the gradient at initialization.*”
>
> **A**: Due to the constancy of NTK [Jacot et al. 2018] and transition to linearity [Liu et al. 2020], the gradient features during and after training keep the same as at initialization. Hence, an analysis at initialization is enough.
>
> [Jacot et al. 2018] Jacot et al. Neural tangent kernel: Convergence and generalization in neural networks, NeurIPS, 2018.
>
> [Liu et al. 2020] Liu et al. On the linearity of large non-linear models: when and why the tangent kernel is constant. NeurIPS, 2020.
>
> **Re**: “*In Theorem 4.2 (non-linear network), only the first layer is trainable, while all subsequent layers remain fixed.*”
>
> **A**: It is often an acceptable setting in theoretical works to fix upper layers to simplify analysis. See for example [Du et al. 2019]. Indeed, we have a section in Appendix F to relax this constraint on upper layers, and similar results also hold. In addition, in our numerical results, all layers are trainable and results are consistent with our theory. Hence, we believe the result should still hold when the constraint is removed.
>
> [Du et al. 2019] Du et al. Gradient Descent Provably Optimizes Over-parameterized Neural Networks, ICLR, 2019.
>
>
> **Re**: “*In fact, for similar data points, increased separation may be undesired, especially when the labels are the same.*” and **Q2**: “*Could you clarify how the separation property of the data points relates to their corresponding labels?*”
>
> **A**: We don’t quite think  increasing separation is undesired when the labels are the same. In fact, for the training data in the same class, whether there is more or less separation among them does not really matter. The classification task is to find decision boundaries between different classes. Therefore, as long as the feature separation is increased for those with different labels so that the decision boundaries need less fine-tuning, the optimization becomes easier.
>
> **Re**: “*The spectral properties of the NTK have been extensively studied in prior work. This manuscript focuses specifically on the condition number of the NTK, which is closely tied to its smallest eigenvalue.*”
>
> **A**: As we also mentioned in the related works, there are extensive studies on NTK spectrum and its smallest eigenvalues. Our work differs from the prior works in the sense that we are the first to investigate the influence of the existence of non-linear activations. We also have the better separation phenomenon (Section 3) which cannot be found anywhere else. Our analysis techniques are also different.
>
> **Re**: “*while the manuscript claims that the condition number is small (line 57), it appears to be quite large when applied to larger datasets*”
>
> **A**: We apologize for not making it crystal clear.  Generally, the condition number is extremely large due to an infinitesimal smallest eigenvalue, and is far greater than the data size $n$. In Line 57, we say it is small, meaning it becomes much smaller. We will make this point clear in the revision.
>
> **Re**: “*Proposition 4.1 has limited applicability, as it is demonstrated only in the context of small datasets where the number of samples n is less than the dimensionality d.*”
>
> **A**: When $n$ is greater than $d$, the smallest eigenvalue of linear NTK becomes zero and its condition number is infinite. The better separation, Theorem 3.2, and better NTK conditioning, Theorem 4.2, still hold.
>
> **Q1**: “*What are the practical implications of your theoretical findings?*”
>
> **A**: We consider our paper as a theoretical work that provides insights and understandings of deep learning. Two possible practical implications include: (1) we may need to rethink the current NTK-based optimization theories which often assume a fixed condition number, our paper suggests the activation function can influence this condition number hence the convergence rate. (2) Another implication of our better separation theorem could relate to the origin of adversarial examples where the adversarial and clean examples can be considered as two similar inputs. Our theory indicates that, due to the non-linearity of activation function, they are quite largely separated in the gradient feature space which may attribute to the observed distinct class predictions between adversarial and clean examples.
>
>
> **Q3**: “*Figure 2 provides numerical evidence that the separation phenomenon also holds for various non-linear activation functions. However, the behaviour differs across ReLU², GeLU, and Tanh. Can your theoretical framework be extended to accommodate other non-linear activations and account for the observed differences?*”
>
> **A**: The main message we had in Figure 2 is that, similar to the ReLU case, for small input angles (i.e., left end of each picture), non-linearly activated NNs are better separated; and when depth increases, the separation is larger. Therefore, the behavior is not significantly different from ReLU.
>
> As for the extension beyond ReLU, the technical difficulty is to carry out a few integrations for other activations (similar to the ones in Line 652 and Line 661). In the paper, we numerically estimated them and showed the results in Figure 2. Therefore, if there exist close-form expressions for these integrations, our theory will extend to those non-ReLU activation functions.

---

> > ### Comment · Reviewer_gxGU · 2025-08-06
> >
> > I have read the authors’ response. Although some issues were addressed, several concerns remain. Specifically, I remain unconvinced by the explanation regarding the effect of separating data points with identical labels. Moreover, the practical implications of the theoretical results could be more clearly demonstrated. Lastly, the theoretical setting appears to be relatively narrow in scope. Nonetheless, my overall evaluation remains positive.

---

> > > ### Author Response · Authors · 2025-08-07
> > >
> > > Thanks for your reply. We hope we can address your remaining concerns by the following:
> > >
> > > **[Data with same label]**: Perhaps think about the analogy: linear model vs. kernel machine (or some feature models with rich feature). In the latter, data are mapped to a rich feature space regardless of their labels; the belief is that the high dimensional feature space makes it easier to separate data with different labels. In this case, whether the data with same label become more separated or not doesn't really matter.
> > >
> > > **[Implications of the theoretical results]**: We believe it is hard to fully explain the ideas of those implications in such a short text. In high level, most prior NTK-based optimization theories obtain convergence rates in terms of NTK or its condition number $\kappa$ [1,2], for example $O(exp(-t/\kappa))$. When comparing these convergence rates, we often implicitly assume that $\kappa$ is only determined by the data and the same function of $\kappa$ means the same rate. Our paper suggests that, $\kappa$ depends on activation function; hence, even when the convergence rate functions have the same form, different model architecture can influence $\kappa$, hence leads to different convergence rates.
> > >
> > > [1]: Du, Lee, Li, Wang, Zhai. Gradient Descent Finds Global Minima of Deep Neural Networks. ICML 2019.
> > >
> > > [2]: Liu, Zhu, Belkin. Loss landscapes and optimization in over-parameterized non-linear systems and neural networks.
> > >
> > > **[narrow scope]**: We believe you meant the theorems are ReLU only. We'd like to point out that our analysis includes other activation functions including ReLU², GeLU and Tanh. With a difficulty of carrying out a few integrations for these activations (which possibly not having close-form expressions fundamentally), we numerically evaluated these integrations and showed their results in Figure 2. The results are consistent with that for ReLU.
> > >
> > > Please let us know if you have further questions.

---

> > > > ### Comment · Reviewer_gxGU · 2025-08-08
> > > >
> > > > Thank you for these clarifications. I believe overall that your paper makes a viable contribution and support its acceptance.

---

### Official Review · Reviewer_L1pt · 2025-06-29

**Clarity:** 3
**Significance:** 3
**Originality:** 3
**Rating:** 5
**Confidence:** 4

**Summary:**

This paper reveals that non-linear activation functions not only enhance expressivity but also improve feature separation and NTK conditioning in wide neural networks. These effects are amplified with network depth, facilitating convergence of gradient-based methods. In contrast, linear networks lack these advantages, preserving the original data geometry and exhibiting poorly conditioned NTKs.  While the theoretical analysis focuses on ReLU, empirical results with other activations support the claims.

**Questions:**

What are the specific technical challenges in extending the theoretical analysis to other activation functions? Is it because, for most non-linear activations, it is difficult to obtain a closed-form expression for the entries of the corresponding NTK? If so, have you considered analyzing the Hermite coefficients? Perhaps the higher-order coefficients could be bounded or even omitted.

**Ethical Concerns:**

["NO or VERY MINOR ethics concerns only"]

**Final Justification:**

The rebuttal by the authors has addressed mostof my concerns. Thus, I raise my score

**Limitations:**

Yes.

**Paper Formatting Concerns:**

There no major formatting issues in this paper.

**Quality:**

3

**Strengths And Weaknesses:**

Strengths

1 This paper is technically sound, The claims are well supported by theoretical analysis on ReLU and by empirical observations on other activations.

2 The paper is well-written and clearly presented, making its theoretical insights and empirical findings easy to follow.

3 The results are important for understanding the role of non-linear activations. The paper provides a comprehensive theoretical analysis demonstrating that the ReLU activation improves  feature separation and NTK conditioning. To the best of my knowledge, these findings are new.

Weaknesses

1 The title should be specific to ReLU activations as the main contribution of this paper is the theoretical findings, which is only conducted on ReLU.

2 This paper demonstrates that as depth increases, the condition number of the NTK matrix with ReLU improves. However, many previous works [1] [2] show that as depth increases, the NTK degenerates. That is, the infinite depth limit of the NTK regime is trivial.

3 To be honest, I acknowledge that the findings of this paper are novel, but not entirely surprising. While the theoretical analysis provides valuable insights, it is limited to ReLU activations. It seems that extending the results to other widely used activation functions may involve technical challenges. If the analysis could be extended to arbitrary non-linear activations, it could offer valuable guidance for the design of new activation functions.

[1] Hayou S, Doucet A, Rousseau J. The curse of depth in kernel regime.

[2] Bietti A, Bach F. Deep equals shallow for ReLU networks in kernel regimes.

---

> ### Author Rebuttal · Authors · 2025-07-30
>
> We thank the reviewer for the comment. We address your concerns and questions below.
>
> **W1**: “*The title should be specific to ReLU activations as the main contribution of this paper is the theoretical findings, which is only conducted on ReLU.*”
>
> **A**: We thank the reviewer for this important suggestion. We will update the title in the revision to include ReLU. Please also note that we present numerical results for other types of activation functions in section 3 (seen in Figure 2).
>
> **W2**: “*This paper demonstrates that as depth increases, the condition number of the NTK matrix with ReLU improves. However, many previous works [1] [2] show that as depth increases, the NTK degenerates. That is, the infinite depth limit of the NTK regime is trivial.*”
>
> **A**: Yes. As our Theorem 4.3 suggests, the NTK becomes trivial for the infinite depth limit, which is consistent with [1,2]. But our main focus is on the finite depth scenario, where NTK and gradient features may not be trivial. At a practical depth, a better feature separation can make classification tasks easier as there is less need to fine-tune the decision boundaries. We focus on how different depths lead to different levels of gradient feature separation and how it affects the NTK matrix.
>
>
> **W3**: “*.... If the analysis could be extended to arbitrary non-linear activations, it could offer valuable guidance for the design of new activation functions.*” and **Q1**: “*What are the specific technical challenges in extending the theoretical analysis to other activation functions?... have you considered analyzing the Hermite coefficients? Perhaps the higher-order coefficients could be bounded or even omitted.*”
>
> **A**: Yes, the technical challenge is to carry out a few integrations for other activations (similar to the ones in Line 652 and Line 661). Instead, we numerically estimated them and showed the results for non-ReLU functions in Figure 2.
>
> We appreciate your valuable suggestion on bounding or omitting higher-order Hermite coefficients. We will try it.

---

> > ### Comment · Reviewer_L1pt · 2025-08-01
> > **Replying to Rebuttal by Authors**
> >
> > I confirm my previous score. The rebuttal by the authors has clarified some of my concerns.

---

### Official Review · Reviewer_sTQk · 2025-06-30

**Clarity:** 3
**Significance:** 3
**Originality:** 3
**Rating:** 4
**Confidence:** 4

**Summary:**

This paper presents simple results on the tangent feature map of deep neural networks: (1) non-linear activations cause the angle in tangent feature space to be larger than input angles, and (2) this effect is amplified with depth. Importantly, this effect is not observed in linear networks. A natural consequence of this increased angle is improved conditioning of the NTK. This phenomenon is also empirically verified on datasets using finite width neural networks.

**Questions:**

1. Can the authors comment on finite width effects?
2. Further, instead of increasing depth, could similar results be derived as a function of width for a fixed depth?

**Ethical Concerns:**

["NO or VERY MINOR ethics concerns only"]

**Final Justification:**

After going through the detailed discussions between the reviewers, AC, and the authors, I will maintain my original score. While there are some concerns regarding scope of settings in which their NTK results hold, I believe that the results in the current state of the paper are significant and novel enough and hence justify my score of 4.

**Limitations:**

Yes, the authors have addressed limitations.

**Quality:**

3

**Strengths And Weaknesses:**

Strengths:

1. The result is simple and presented in a straightforward and easy to follow manner.
2. It provides a novel perspective and concrete explanation (as far as this reviewer is aware) for the utility of non-linear activation functions.
3. Theoretical results seem sound and are numerically verified as well.

Weaknesses:

1. The theory depends on infinite width of the network, and is restricted to ReLU activations.
2. The section on optimization acceleration could benefit from more quantitative analysis. For example, a quantitative convergence improvement as a function of depth, informed by the main result would be insightful. Especially if this quantitative analysis is corroborated by the increasing depth experiment that is presented. Currently, the faster convergence for higher depths is qualitative and does not build on the main result in a theoretical capacity.
3. Some of the writing could be slightly improved

---

> ### Author Rebuttal · Authors · 2025-07-30
>
> We thank the reviewer for the comment. We address your concerns and questions below.
>
> **W1**: “*The theory depends on infinite width of the network, and is restricted to ReLU activations.*” and **Q1**: “*Can the authors comment on finite width effects?*”
>
> **A**: Our theory is conducted in a sufficiently large but finite width setting (See for example Theorem 3.2). Basically, our theory is based on the well-established large-width NN analysis (such as [8,9,15,17,19] in the reference list of our paper), where the NN deviates from an infinitely wide NN slightly (in the order of $O(1/\sqrt{m})$). While there is not much theory for smaller network width, we temporarily only consider this  sufficiently large but finite width setting.
>
> In fact, we considered other activation functions including ReLU², GeLU and Tanh (See Figure 2 and its discussion). Numerical evaluations show that similar results hold for these activation functions. The difficulty for theoretical analysis of these non-ReLU activations is to carry out the integrations similar to those in Line 652 and Line 661. Nevertheless, these integrals can be estimated numerically.
>
>
> **W2:** “*The section on optimization acceleration could benefit from more quantitative analysis. For example, a quantitative convergence improvement as a function of depth, informed by the main result would be insightful. … Currently, the faster convergence for higher depths is qualitative and does not build on the main result in a theoretical capacity.*”
>
> **A**: We agree that a quantitative result on the optimization acceleration would be more interesting. While we can get a quantitative result in the simple setting of data size equal $2$ based on the quantitative result of feature separation – Corollary 3.3, the major difficulty for general case is that the connection between the smallest eigenvalue of NTK and data feature separation is still not quantitatively clear, therefore, the quantitative result of feature separation downgrades to a qualitative result of NTK. We conjecture that a sophisticated analysis involving random matrix theories can achieve this goal. We consider it as a future work.
>
>
> **Q2**: “*Further, instead of increasing depth, could similar results be derived as a function of width for a fixed depth?*”
>
> **A**: Lemma 3.1 and Theorem 3.2 show that, with a fixed depth, the width contributes minimally compared to a width-independent term (see for example Eq. (7)). Our paper currently focuses on the leading term (i.e., the width-independent term). As for the width-*dependent* term, we believe it also depends on the initialization seed, in which case we only obtain an upper bound.

---

> > ### Comment · Reviewer_sTQk · 2025-08-06
> > **Response**
> >
> > Thank you for your comments.
> >
> > Sure, it is technically finite-width but the analysis is still the NTK regime, i.e., the "close to linear model" regime. The point you bring up on numerical integrations is interesting, and I would be curious to see some results along these lines. It would add to the scope of the paper if these results somewhat align with the NTK regime theory.
> >
> > Thank you for the comment on the activation functions, this addresses my concerns.
> >
> > Thank you for the comment on W2. The random matrix analyses would greatly benefit the strength of this paper, but in its current state I shall maintain my score.
> >
> > Thank you for pointing out the width-dependent term in Thm 3.2. This addresses my main question well.
> >
> > Overall, my concerns have been addressed, but due to the scope of some of the theoretical results still being open, I will maintain my score.

---

> > > ### Author Response · Authors · 2025-08-07
> > >
> > > Thanks for your detailed reply. We are glad to see your concerns are addressed.
> > >
> > > As for the scope of the theoretical results, we would like to further clarify below:
> > >
> > > [Regarding NTK regime]: we note that most well-developed theories of deep learning are still based in the NTK regime, and there are not much theory beyond. As a theoretical paper that is based on previous theories, we believe it is acceptable to focus on this regime at this moment. Please note that our experiments are not in the NTK regime, since the network width is much smaller than the NTK theory requires, and are consistent with the NTK-regime-based theory.
> > >
> > > [quantitative analysis]: We'd like to point out that, although the condition number result need more random matrix analyses to be quantitative, the **better separation results, Corollary 3.3 and Theorem 3.5, are quantitative**, novel and significant enough.

---

### Official Review · Reviewer_p5SR · 2025-06-30

**Clarity:** 4
**Significance:** 3
**Originality:** 3
**Rating:** 5
**Confidence:** 5

**Summary:**

This paper studies the effect of non-linear activation functions on the NTK and its associated feature map $x \mapsto \nabla_\theta f_\theta (x)$ in infinite-width neural networks.
Based on the fact that feature norms are (virtually) unchanged by linear and ReLU networks, the authors focus on the study of the separation angle between data points and between their features.

They show that using linear activation functions results in a Gram matrix which is (virtually) unchanged across the hidden layers: the angles between data do not change and the associated NTK has a condition number that corresponds to the Gram matrix of the original data. Conversly, the use of activation functions increases the angles between data with small angles, and improves the NTK condition number.

More specifically, for ReLU networks, the paper provides a recursion relation for the angle between data points. This recursion allows the authors to study the infinite-width and infinite-depth networks, showing that:
- All data pairs have an angle that converges to $\sim 75,5$ degrees.
- When the data lie on a sphere, the condition number will converge to $\frac{n+4}{3}$.
- The condition number decreases with the depth of the network.

The connection between the condition number and the training speed allows them to discuss implications for convergence speed.  Various experiments support these theoretical findings.

**Questions:**

- Fast convergence  can occur with a decrease of generalization; for instance, a NTK Gram matrix equal to the identity matrix would lead to fast convergence but no generalization capability. Could you discuss this trade-off explicitly? Specifically, how does generalization behave in Figure 4 (the experiments that illustrate the increase of training speed) ?
- Relatedly, the experiments results provide an interesting fact that is not discussed in the paper: in the ReLU setting, angles converge to around 75.5 degrees, whereas with other activation functions used in Figure 2, it seems that the angles converges to around 90 degrees, and this could potentially reduce generalization properties. Could you provide the generalisation for these activation function in the setup of Figure 2 and discuss about this depending of what you obtain? Also can you report the limiting angles somewhere?
- In corollary 3.3, I suggest you could explicitly mention an insightfull consequence: $L$ should be of order $\frac{1}{\theta_{in}}$ for $\Phi(x,z)$ to be non trivial. Do you agree with this statement?
- Proof clarifications:
    - Isn’t Lemma 4.1 a consequence of the Law of Large Numbers, given that each coordinates is the average of m independent variables?
    - While the result is correct, and equation (40) captures the intuition, I believe it’s inaccurate to claim “solving the above equation (40), we found that …”. One should keep all approximations orders and prove that the l.h.s. is of order $\frac{1}{L^2}$, thus implying that $1-4\cos(\Phi_L) \to 0$ as $L\to\infty$. Do you agree ?

**Ethical Concerns:**

["NO or VERY MINOR ethics concerns only"]

**Final Justification:**

The authors have addressed all my points, in particular those about generalization, relation with previous works, and certain technical details. After reading the other reviews and the authors’ responses, I confirm my score.

**Limitations:**

Yes

**Quality:**

3

**Strengths And Weaknesses:**

Strengths:
- The paper is clearly written and easy to follow, with an equality well-written and well-organized appendix.
- It provides a fairly comprehensive study of the effect of ReLU activation function  on the NTK with no bias terms. A clear study of these neural network which lies at the boundary between the order and freeze phase was to my knowledge lacking. Its investigates both the separation angles and the features norms, provides recursive relations on angles, analyzes limits and variation of the condition number. The proofs feature interesting random matrices and geometric flavours, taking full advantage of the specific activation function they consider, the ReLU. I particularly like the result concerning the limiting angle,  and whose value is non-trivial (i.e. neither 0 nor 90 degrees).  I think this is one of the main new contributions of the paper, along with the decreasing property of the condition number and its explicit limiting value. The paper thus provides interesting quantitative results on top of the qualitative results.
- Empicial results show that these theoretical insights can be extended to other activation functions.



Weaknesses:
- Link with previous literature is incomplete:
    - Reference [1] shows that a transition occurs for the NTK between a “freeze” (order) phase, where the scaled NTK converges to a constant, and a “chaos” phase where the NTK looks more like a Dirac Kernel. It was shown that, in term of their characteristic value, ReLU networks without bias lie on the boundary between order and chaos. The current paper extends the study of this phenomena by looking specifically at what happens at the boundary, but does not explicitly mention this connection.
    - I believe that the linear NTK has been studied before or is classical knowledge (it results directly from the recurrence relation in Theorem 1 of [2]). Check if it was not provided somewhere else.
    - The improved conditioning property (though without quantitative results provided in the paper) between linear and non-linear activation were already known. Specifically, the linear NTK has infinite condition number when the number of data points exceeds the dimension. Whereas, as shown in  Appendix A.4 [2], the NTK restricted to the sphere is positive-definite as soon as the activation function is non-polynomial.
    - Could you discuss also relation with [3].

- It would be interesting to explicitly discuss the role of bias, as they can significantly affect the behaviour of the NTK.
- The paper focuses exclusively on the speed of convergence, and does not discuss the generalization performance, despite a trade-off between these two effects. Empirical results exploring generalisation behaviour would be give valuable additional insights.



[1] - Freeze and Chaos: NTK views on DNN Normalization, Checkerboard and Boundary Artifacts, Jacot, Gabriel, Ged, Hongler (https://proceedings.mlr.press/v190/jacot22a/jacot22a.pdf)

[2] - Neural Tangent Kernel: Convergence and Generalization in Neural Networks, Jacot, Gabriel, Hongler, ArXiv Version https://arxiv.org/pdf/1806.07572

[3] - Bounds for the smallest eigenvalue of the NTK for arbitrary spherical data of arbitrary dimension Karhadkar, Murray, Montúfar https://arxiv.org/abs/2405.14630

---

> ### Author Rebuttal · Authors · 2025-07-30
>
> We thank the reviewer for the insightful and detailed comments. We’d like to address your concerns and questions below.
>
> **W1a**: “*Reference [1] shows that a transition occurs for the NTK between a “freeze” (order) phase ... and a “chaos” phase. The current paper … looking specifically at what happens at the boundary, but does not explicitly mention this connection.*”
>
> **A**: We thank the reviewer for providing this related work. We will cite it in the revision and clarify this connection. Basically, our theory only considered the bias-free setting, which is widely adopted by many prior theoretical works, such as [Du et al. 2019]. After reading [1], we agree that the bias makes a big difference (bigger than we previously thought) and it is interesting to extend the analysis to include the bias term. We also realize that this extension is technically non-trivial (just like observed in [1], different scaling $\beta$ of the bias leads to distinct results for NTK). We would like to consider it as a future work.
>
> [Du et al. 2019]: Du et al. Gradient Descent Finds Global Minima of Deep Neural Networks. ICML 2019.
>
> **W1b**: “*I believe that the linear NTK has been studied before or is classical knowledge (it results directly from the recurrence relation in Theorem 1 of [2]).*”
>
> **A**: While the linear NTK result, i.e., Corollary 2.3, can be a direct consequence of Theorem 1 of [2], the feature angle separation theorem, Theorem 2.2, does not follow [2]. We didn’t find any prior work having the result of Theorem 2.2, although it is not technically hard. Hence, we put it in the Preliminary section and do not claim it as a contribution. In addition, we will clarify the connection of Corollary 2.3 and Theorem 1 of [2] in the revision.
>
> **W1c**: “*The improved conditioning property…between linear and non-linear activation were already known. Specifically, the linear NTK has infinite condition number when the number of data points exceeds the dimension. Whereas … the NTK restricted to the sphere is positive-definite as soon as the activation function is non-polynomial.*”
>
> **A**: The known result is only for the special case of an infinite linear NTK condition number. In addition, it also requires the data lies in a unit sphere. Our result not only applies for more general settings (e.g., linear NTK is also positive-definite), but, more importantly, focuses on the better separation of gradient features, which is not known from prior works. We’d like to cite [2] and add a discussion about the connection.
>
> **W1d**: “*Could you discuss also relation with [3].*”
>
> **A**: The paper [3] provides both upper and lower bounds for the smallest eigenvalues of NTKs for both shallow and deep NN. First, while these bounds are insightful and valuable, it is different from our results as it is not clear in [3] what effect the activation function has on the smallest eigenvalue. In addition, [3] obtained a linearly increasing upper bound for the  smallest eigenvalue: $\lambda_{min}< L$. However, note that when depth $L$ increases, NTK accumulates (an addition of $L$ positive definite terms), also resulting in an linear increasing top eigenvalue, hence condition number is still independent of depth $L$. Therefore, we also don’t see a depth-dependent NTK condition number in [3], which is also different from ours.
> Moreover, we conjecture that the bounds obtained in [3] is not tight enough to extract the effect of non-linear activations and depth. As we argued above, upper bounds of [3] would leads to a depth independent bounds for condition number, which hides the depth dependent nature of the true value of condition number, due to a upper bound that is not tight enough.
>
> **W2**: “*It would be interesting to explicitly discuss the role of bias, as they can significantly affect the behaviour of the NTK.*”
>
> **A**: Yes, we agree that it is interesting to discuss the role of bias. With a quick trial, we found that the analysis with different scaling $\beta$ of the bias is technically non-trivial. Similar to the distinct results observed in [2] for different $\beta$, we believe the analysis of bias could potentially lead to new observations, which will be quite interesting. We consider it as a future work.
>
> **W3**: “*The paper … does not discuss the generalization performance... Empirical results exploring generalisation behaviour would be give valuable additional insights.” and  Q1: “Fast convergence can occur with a decrease of generalization…. Could you discuss this trade-off explicitly? Specifically, how does generalization behave in Figure 4 (the experiments that illustrate the increase of training speed) ?*”
>
> **A:** We agree there is often a trade-off between optimization and generalization, especially when depth is large. We will add discussions about this trade-off in the revision.
>
> As our theory suggests in Theorem 3.5 and Theorem 4.3, in the extreme case of infinite depth, any input pairs become equally separated in gradient features regardless of their original similarity. Even though not mutually orthogonal, this could also result in a trivial generalization: close to random guess for unseen data. This somehow can also be obtained from [Wang et al. 2020], where they dropped the initial random guess value and obtained a zero prediction for unseen data. We will cite this paper in the revision.
>
> As for finite depth, there is no theory that clearly suggests at what depth this trade-off starts to happen. In our experiments in Figure 4, we can slightly see this trade-off. For example, for MNIST, the test accuracies for different depths $L$ are:
>
>      95.98% (L=1),  97.43% (L=3),  97.57% (L=6),  97.52% (L=8),  97.39% (L=10),  97.19% (L=12)
>
> Above, we can see a decreased generalization starting from a certain depth (in this case around $L=8$).
>
> [Wang et al. 2020]: Wang et al. Why Do Deep Residual Networks Generalize Better than Deep Feedforward Networks? — A Neural Tangent Kernel Perspective. NeurIPS. 2020.
>
>
> **Q2**: “*... in the ReLU setting, angles converge to around 75.5 degrees, whereas with other activation functions used in Figure 2, it seems that the angles converges to around 90 degrees, and this could potentially reduce generalization properties. Could you provide the generalisation for these activation functions in the setup of Figure 2 and discuss this depending on what you obtain? Also can you report the limiting angles somewhere?*”
>
> **A**: As mentioned above, very deep ReLU NNs already have a reduced generalization performance and converge to random guess at infinite depth, hence we don’t think the difference between $75.5$ degree for ReLU and $90$ degree for others make much difference in generalization.
>
> We will also add the (numerical-based) observations about the limiting angles of other activation functions in the revision.
>
>
>
> **Q3**: “*In corollary 3.3, I suggest you could explicitly mention an insightful consequence: $L$ should be of order $1/\theta_{in}$ for $\phi$ to be non trivial. Do you agree with this statement?*”
>
> **A**: Yes. We agree. We will update it in the revision.
>
>
> **Q4-1**: “*Isn’t Lemma 4.1 a consequence of the Law of Large Numbers …*”
>
> **A**: There is no Lemma 4.1 in our submission. We’d like to ask which lemma you refer to. We are happy to make further clarification after.
>
> **Q4-2**: “*I believe it’s inaccurate to claim ‘solving the above equation (40), we found that …’  …Do you agree?*”
>
> **A**: Thanks for pointing out this issue. Yes, you are correct. We will update our proof in the revision. Basically, based on Eq. (40), we can see that $1/4$ is an attraction point for the series $\cos \phi_L$ (look at the sign of $Delta \cos \phi_L$). With a bit of effort we can show that the series converges. Then, suppose $\cos \phi_L$ converges to a point not equal to $1/4$, the l.h.s. of Eq, (40) will be at the order of $1/L$, which results in a non-converging $\cos \phi_L$, contradictory. Hence, we can show that the series converges to $1/4$.

---

> > ### Comment · Reviewer_p5SR · 2025-08-05
> >
> > Thank you for this detailed response. Regarding Lemma 4.1, I am sorry, I was indeed referring to Lemma D.1. I think this would shorten the proof.
> >
> > The authors have addressed all the comments raised. After also reading the other reviews and rebuttals, I confirm the score.

---

> > > ### Author Response · Authors · 2025-08-07
> > >
> > > Thank you for your reply. We appreciate your suggestion for the proof of Lemma D.1, we will revise it.
> > >
> > > We are glad that all your concerns are satisfactorily addressed. We remain available for any further questions you may have.

---

### Official Review · Reviewer_ndzW · 2025-07-03

**Clarity:** 3
**Significance:** 2
**Originality:** 3
**Rating:** 4
**Confidence:** 3

**Summary:**

The paper discusses properties of neural tangent kernels and the effects of non-linear activation functions compared to linear activations. The field of discussion lies in the application to hyper-wide neural networks and examines the relation between network depth and width and the angular separation of internal representations in model gradient space. The key findings suggest that increasing the network depth significantly enhances the feature separation observed in wide models with non-linear activations. In particular, the authors show that in the infinite-width-then-depth limit, all data are equally separated with an angle ∼75.5 degrees.

The authors support their claims with detailed mathematical analysis under standard assumptions for wide network limits (e.g., random initialization, asymptotic regimes). They also validate these insights empirically using synthetic and real datasets like MNIST and LibriSpeech. Their findings offer a new perspective on why non-linear activations are crucial for effective learning in deep models.

**Questions:**

While ReLU is studied in depth, can the theoretical framework be extended to other activation functions and their derivatives? Would the conclusions (and numerical limits) still hold? If not, what aspects of the analysis break down?

What are the practical takeaways for model design or training procedures, how your findings can be used in certain settings?

**Ethical Concerns:**

["NO or VERY MINOR ethics concerns only"]

**Final Justification:**

The paper contains interesting results though for a limited scenario. During the rebuttal authors addressed the challenges of generalizing theory, and taking into account they did experiments to empirically observe  similar behavior on other activations, I put my score as slightly above the borderline.

**Limitations:**

Yes

**Paper Formatting Concerns:**

No major issues

**Quality:**

3

**Strengths And Weaknesses:**

The paper presents solid theoretical analysis with lemmas and theorems grounded in random matrix theory and kernel methods. Experimental results align well with theoretical predictions and are reproducible via detailed descriptions in Section 4.1.
The paper is well-written, logically structured, and accessible to readers familiar with NTK and wide network theory. The claims made in the abstract and introduction match the contributions in the body of the paper.

The insights into how non-linearities improve optimization dynamics via NTK conditioning and representation learning via feature separation are valuable for understanding modern current DL systems. While not  proposing a novel method, it offers interpretability tool and define specific numerical limits for parameter spaces.

Focus solely on ReLU activations without considering a broad array of possible functions around may miss some insights.

---

> ### Author Rebuttal · Authors · 2025-07-30
>
> We thank the reviewer for the comment. We address your concerns and questions below.
>
> **W1** : “*Focus solely on ReLU activations without considering a broad array of possible functions around may miss some insights.*” and **Q1**: “*... can the theoretical framework be extended to other activation functions and their derivatives? Would the conclusions (and numerical limits) still hold?*"
>
> **A**: In fact, we considered other activation functions including ReLU², GeLU and Tanh (See Figure 2 and discussions at Line 238). Numerical evaluations show that similar results hold for these activation functions. We believe the results holds more generally. The difficulty for theoretical analysis of these non-ReLU activations is to carry out the integrations similar to those in Line 652 and Line 661. Nevertheless, these integrals can be estimated numerically, as we did in Figure 2.
>
>
> **Q2**: “*What are the practical takeaways for model design or training procedures, how can your findings be used in certain settings?*”
>
> **A**: We consider our paper as a theoretical work that provides insights and understandings of deep learning. Two possible practical implications include: (1) we may need to rethink the current NTK-based optimization theories which often assume a fixed condition number, our paper suggests the activation function can influence this condition number hence the convergence rate. (2) Another implication of our better separation theorem could relate to the origin of adversarial examples where the adversarial and clean examples can be considered as two similar inputs. Our theory indicates that, due to the activation function, they are quite largely separated in the gradient feature space which may attribute to the observed distinct predictions between adversarial and clean examples.

---

> ### Author Response · Authors · 2025-08-07
>
> Dear Reviewer ndzW:
>
>    We would like to reach out to see if your concerns have been resolved or whether you have further questions. We are happy to address any remaining concerns/questions. If no further concerns, we kindly ask the reviewer to consider updating their overall rating. Thank you very much!

---

> > ### Comment · Reviewer_ndzW · 2025-08-07
> >
> > Dear Authors,
> >
> > Thank you for your reply, my concerns regarding activation functions have been addressed, also answers to other reviewers provided good clarification, I  intend to raise the score and would be curious to see the mentioned implications explored in future works.
> >
> > Best,
> > Reviewer

---

> > > ### Comment · Reviewer_ndzW · 2025-08-08
> > >
> > > Dear Authors,
> > >
> > > Thanks again for outlining the possible implications for optimization and adversarial robustness in your response. I understand that the results are theoretical and derived under assumptions (at initialization, infinite-width limit, linearized regime). I’m curious whether you see any path for examining whether similar patterns emerge during training. And I’d be interested in your thoughts on which assumptions in the current setting might be relaxed in future work to move closer to practical models.
> > >
> > > Best,
> > > Reviewer

---

> > > > ### Author Response · Authors · 2025-08-08
> > > >
> > > > Thank you very much for your quick response and intention to raise the score!
> > > >
> > > > Yes, we believe similar results still hold during training, one possible way is to experimentally check these results during training, we plan to report such results in the revision. Thanks for your suggestion.
> > > >
> > > > As for the assumptions to relax, we discuss them below:
> > > >
> > > > [Infinite-width] We think this can be relaxed once a quantitative result for Theorem 4.2 is obtained, which will enable tolerance to fluctuations of NTK happening in finite-width cases (randomness of the initialization give rise to fluctuations/uncertainty of NTK at finite width). As we mentioned in the reply for other reviewers, such a quantitative result needs sophisticated random matrix analysis.
> > > >
> > > > [at initialization] In the infinite-width case, the results naturally extend beyond initialization, as the model gradient features and NTK are unchanged during training. For finite-width, this relies on analysis of how NTK changes during training, which we haven't been aware of much such analysis in literature. But we believe a one-step analysis is possible, given a rich literature on the one-step (after initialization) analysis.
> > > >
> > > > [NTK regime] There are a lot of experimental work, but little theory,  beyond the NTK regime. Hence, we think experimentally check our results beyond the NTK regime should be a good approach.
> > > >
> > > > Lastly, we'd like to gently remind the reviewer to update the score. Thanks again. We appreciate your constructive comments and suggestions.

---

> > > > > ### Comment · Reviewer_ndzW · 2025-08-09
> > > > >
> > > > > Dear Authors,
> > > > >
> > > > > Thank you for detailed reply and clarifying challenges related to relaxation of the current settings.
> > > > >
> > > > > As far as I'm concerned the score change in the form should be done when submitting final recommendations after Reviewers-AC discussion
> > > > >
> > > > > Best,
> > > > > Reviewer

---

> > > > > > ### Author Response · Authors · 2025-08-09
> > > > > >
> > > > > > Thank you very much for your response! We are glad it clarifies!
> > > > > >
> > > > > > We totally understand your intention of holding the score change until Reviewers-AC discussion. While we are not aware of such an official "final recommendation" step in NeurIPS (we could be wrong, we apologize), we believe during Reviewers-AC discussion it is allowed.
> > > > > >
> > > > > > Best,
> > > > > > Authors

---

### Public Comment · ~Dávid_Terjék1 · 2025-11-29
**All results can be found in earlier preprints**

I would like to bring to the attention of the authors and the reviewers that all theoretical results in this work have been previously available (in greater generality) in the following two preprints, uploaded to arXiv 10 months ago:

[1] MLPs at the EOC: Spectrum of the NTK by D. Terjék, D. González-Sánchez (https://arxiv.org/abs/2501.13225)

[2] MLPs at the EOC: Concentration of the NTK by D. Terjék, D. González-Sánchez (https://arxiv.org/abs/2501.14724)

In particular,
 - Lemma 3.1 is implied by Proposition 10 of [1],
 - Theorem 2.3, Corollary 3.3 and Theorem 3.5 are implied by Proposition 13 and Proposition 14 of [1],
 - Corollary 2.4, Theorem 4.2 and Theorem 4.3 are implied by Theorem 18 of [1] and
 - Theorem 3.2 is implied by Theorem 28 of [2] together with Proposition 13 and Proposition 14 of [1]. (Note that Theorem 28 of [2] gives a bound for the NTK matrix, but it follows from an entrywise concentration bound that can be found in the proof.)

The papers [1] and [2] prove these results and more for MLPs in both the kernel and rich regimes for arbitrary homogeneous activation functions and hidden layer widths, doing so in a way that quantifies the effects such hyperparameters have on the initial NTK.

---

### Note · Authors · 2025-08-15

We thank the AC and all reviewers for their active engagement and constructive suggestions. We are delighted that the discussion has satisfactorily addressed all concerns and questions and all reviewers vote for acceptance according to their response. We promise to include your suggestions in revision.

---

### Decision · Program_Chairs · 2025-09-17

**Decision:**

Accept (poster)

**Comment:**

This paper discovers that non-linear activations, compared with linear networks, lead to larger feature separation in the gradient feature space and improved NTK conditioning in the infinite-width limit. The theoretical analysis focuses on ReLU while empirical observations are consistent across other activations.

The reviewers acknowledge that the discovery is simple (in the good sense), novel and insightful, supported by rigorous and non-trivial analysis, with interesting quantitative results. The core contributions are clearly presented with high-quality formulations, statements, and informative experiments. The paper is easy to read despite the theoretical depth involved.

The current analysis is restricted to ReLU and the NTK regime, which limits the generality of the formal results. This should be better reflected in the title and/or abstract. While the discovery is interesting, its implications for applications --- including how practitioners could benefit from the better separation --- remain unclear.

All five reviewers recommend acceptance with high confidence, and so does the AC.